# Probing protein flexibility reveals a mechanism for selective promiscuity

Nicolas A Pabon, Carlos J Camacho*

Department of Computational and Systems Biology, University of Pittsburgh, Pittsburgh, United States

**Abstract** Many eukaryotic regulatory proteins adopt distinct bound and unbound conformations, and use this structural flexibility to bind specifically to multiple partners. However, we lack an understanding of how an interface can select some ligands, but not others. Here, we present a molecular dynamics approach to identify and quantitatively evaluate the interactions responsible for this selective promiscuity. We apply this approach to the anticancer target PD-1 and its ligands PD-L1 and PD-L2. We discover that while unbound PD-1 exhibits a hard-to-drug hydrophilic interface, conserved specific triggers encoded in the cognate ligands activate a promiscuous binding pathway that reveals a flexible hydrophobic binding cavity. Specificity is then established by additional contacts that stabilize the PD-1 cavity into distinct bound-like modes. Collectively, our studies provide insight into the structural basis and evolution of multiple binding partners, and also suggest a biophysical approach to exploit innate binding pathways to drug seemingly undruggable targets.

*For correspondence:
ccamacho@pitt.edu

**Competing interests:** The authors declare that no competing interests exist.

## Introduction

Structural and proteomic research over the past decade has supplanted the traditional structure-function paradigm by establishing the functional relevance of protein dynamics (*Wright and Dyson, 1999*; *Dunker et al., 2000*; *Haynes et al., 2006*; *Romero et al., 2006*; *Ward et al., 2004*; *Xie et al., 2007*). In particular, eukaryotic regulatory and signaling proteins are skewed toward notably higher degrees of flexibility when compared to other functional categories (*Liu et al., 2009*; *Iakoucheva et al., 2002*). Regulatory proteins also tend toward comparatively higher degrees of binding promiscuity, and we have previously shown thermodynamically how the entropy associated with their flexibility can relate to their specificity toward multiple binding partners (*Liu et al., 2009*). However, a structural understanding of how this selective promiscuity is achieved is still lacking.

Flexible human regulatory proteins such as MDM2 and PD-1 usually only crystallize when ligand-bound. Although nuclear magnetic resonance (NMR) can occasionally resolve unbound (apo) structures of these proteins, it is noteworthy that their apo NMR ensembles often deviate from their bound crystal structures (*Schon et al., 2002*; *Lo Conte et al., 1999*; *Betts and Sternberg, 1999*; *Cheng et al., 2013*; *Zak et al., 2015*; *Lázár-Molnár et al., 2008*; *Lin et al., 2008*). Thus, for many such proteins, available structural data do not capture the full binding dynamics, and the pathway from the apo, non-bound-like state to the bound-like state is unclear. This lack of data obscures the mechanistic connection between interface flexibility, binding promiscuity, and ligand specificity. Moreover, given that many regulatory proteins are promising drug targets, this missing puzzle piece often spells failure for drug design efforts that only target the bound-like state, assuming that this state is available in the apo ensemble. Rational approaches to target flexible proteins will thus benefit from new methods that can reveal the binding pathways connecting the non-bound-like to the bound-like states.

**eLife digest** Many proteins need to interact with other proteins to carry out their various tasks in cells. Such interactions are essential for almost all biological processes and are often disrupted in disease. Cells have thousands of different types of proteins and each has a unique shape that determines which other proteins it can bind to.

It was previously thought that two proteins bind to each other in a manner similar to that of a lock and a key, in which the rigid shape of one protein meshes perfectly with the rigid shape of its partner. However, many proteins are flexible and adopt different shapes depending on whether they are attached to their partner, or not. Moreover, an individual protein may also bind to several different partners, each requiring that protein to adopt several different shapes. These observations have challenged the lock and key model and suggest that flexibility in the structure of a protein plays a key role in its binding to other proteins. However, it is not clear how structural flexibility enables a protein to bind to several different partners while being selective enough to prevent the protein from binding to the wrong ones.

A protein called PD-1 is involved in immune responses in humans and is an emerging target for drugs to treat cancer. Pabon and Camacho used computer simulations to model PD-1's structural flexibility and to find out how this enables the protein to form different shapes when it binds to different partners. The experiments show that the region of PD-1 that binds to other proteins adopts a different shape in the absence and presence of its partners. The binding partners make initial contact with PD-1 via specific features that they share in common. This causes PD-1 to change shape, uncovering a surface of PD-1 that is flexible and is able to accommodate a variety of partners. After this, the binding partners form additional contacts with PD-1 that are specific to each partner.

These findings suggest that the ability of a protein to bind to several different partners is unlocked by certain structures that are present in the binding partners. These structures are found in proteins produced by many different organisms, suggesting that this mechanism is likely to be widespread in nature. This work may open up new avenues for designing drugs to target PD-1 and other proteins that contribute to disease but have so far been impossible to target with drugs.

Binding to flexible receptors is traditionally described by conformational selection (*Ma et al., 1999*; *Tsai et al., 1999*) or induced fit (*Koshland, 1958*) mechanisms, and NMR techniques are often used to distinguish between these two (*Figure 1*). Generally speaking, one assumes a conformational selection scenario if the apo protein ensemble samples bound-like states (apo$_{BL}$) (*Boehr et al., 2009*; *Hoang and Prosser, 2014*). If not, one assumes induced fit (*Schon et al., 2002*). In reality, whether a protein-protein interaction occurs via conformational selection or induced fit depends on the flux of the system through the two alternate pathways from the non-bound-like apo state (apo$_{NBL}$) to the bound-like encounter complex (EC$_{BL}$) (*Hammes et al., 2009*). Flux through the conformational selection pathway is limited by the free-energy difference between the apo$_{BL}$ and apo$_{NBL}$ states, $\Delta G_{BL}^{apo}$, which determines the fractional population of the bound-like state and thus restricts when selection-association with the ligand can occur. On the other hand, flux through the induced fit pathway is for the most part independent of $\Delta G_{BL}^{apo}$, as the ligand is presumed to be able to associate with all apo receptor microstates. Instead, flux through this pathway is limited by the free-energy difference between the EC$_{BL}$ and the non-bound-like encounter complex (EC$_{NBL}$), $\Delta G_{BL}^{EC}$, which is a function of specific interactions between receptor and ligand, and the energy barrier between these states. Both pathways terminate via a ubiquitous optimization step in which minor structural rearrangements at the EC$_{BL}$ interface lead to the high-affinity complex.

To shed light on the structural basis of selective promiscuity in the aforementioned class of flexible-interface multi-ligand proteins, we study the binding mechanism of PD-1 to its cognate ligands PD-L1 and PD-L2. Human PD-1 is a T-cell receptor and immune response regulator that has recently emerged as a breakthrough anticancer target (*Dömling and Holak, 2014*; *Couzin-Frankel, 2013*). NMR and crystallographic studies have revealed the flexibility of the PD-1 interface by showing that its apo and bound conformations are very different (*Cheng et al., 2013*; *Zak et al., 2015*; *Lázár-*

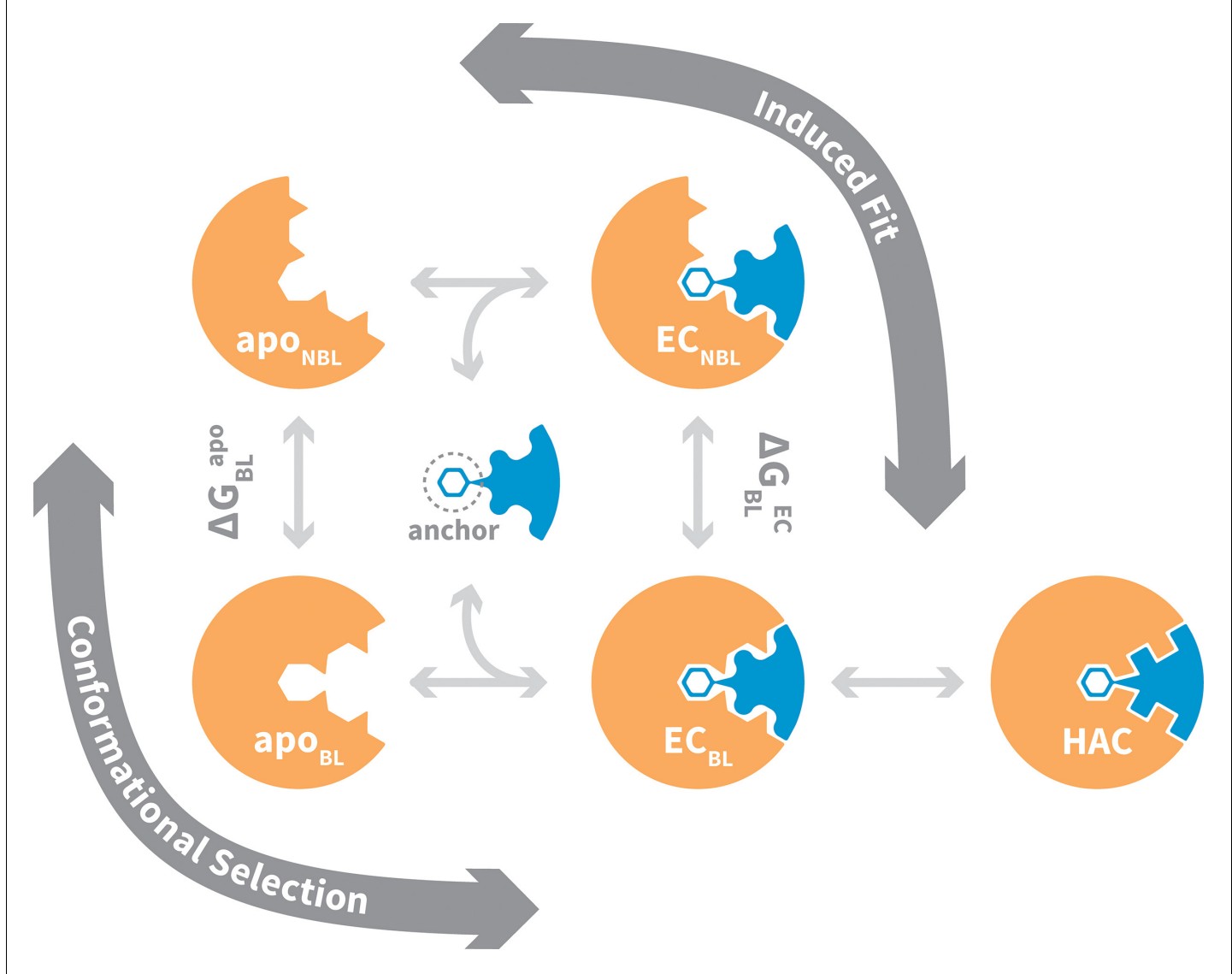

**Figure 1.** General mechanism for ligand binding to flexible receptor. In the conformational selection pathway, the ligand docks to the bound-like (BL) form of the apo receptor (apo$_{BL}$) to form the bound-like encounter complex (EC$_{BL}$). In the induced fit pathway, the ligand docks to the non-bound-like (NBL) form of the apo receptor (apo$_{NBL}$) to form the non-bound-like encounter complex (EC$_{NBL}$). Intermolecular interactions then drive structural transitions to the EC$_{BL}$. Both pathways end with a final induced fit step that optimizes interface side chains, transitioning to the high-affinity complex (HAC). The binding mechanism also highlights an anchor residue often found to be important in molecular recognition (*Rajamani et al., 2004*).

*Molnár et al., 2008*; *Lin et al., 2008*) (*Figure 2*, *Figure 2—figure supplement 1*), suggestive of an induced fit mechanism. Specifically, while the apo PD-1 interface shows a polar surface around Asn66 with an unmatched NH2 (*Figure 2a*), in complex this NH2 group forms two hydrogen bonds, with the PD-L1–bound interface exhibiting a hydrophobic patch around Ile126 (*Figure 2b*), and the PD-L2–bound interface forming a large hydrophobic cavity flanked by Ile126 and Ile134 (*Figure 2c*, *Figure 2—figure supplement 2*).

To date, no small molecular weight PD-1 inhibitors have been reported in the literature despite the importance of this blockbuster target (*Dömling and Holak, 2014*; *Couzin-Frankel, 2013*; *Zarganes-Tzitzikas et al., 2016*). This was somewhat surprising, since the Trp110 binding site observed in the PD-L2–bound cocrystal (*Figure 2c*) displays two key characteristics known to be favorable for ligand binding: concavity (*Laskowski et al., 1996*; *Liang et al., 1998*) and hydrophobicity

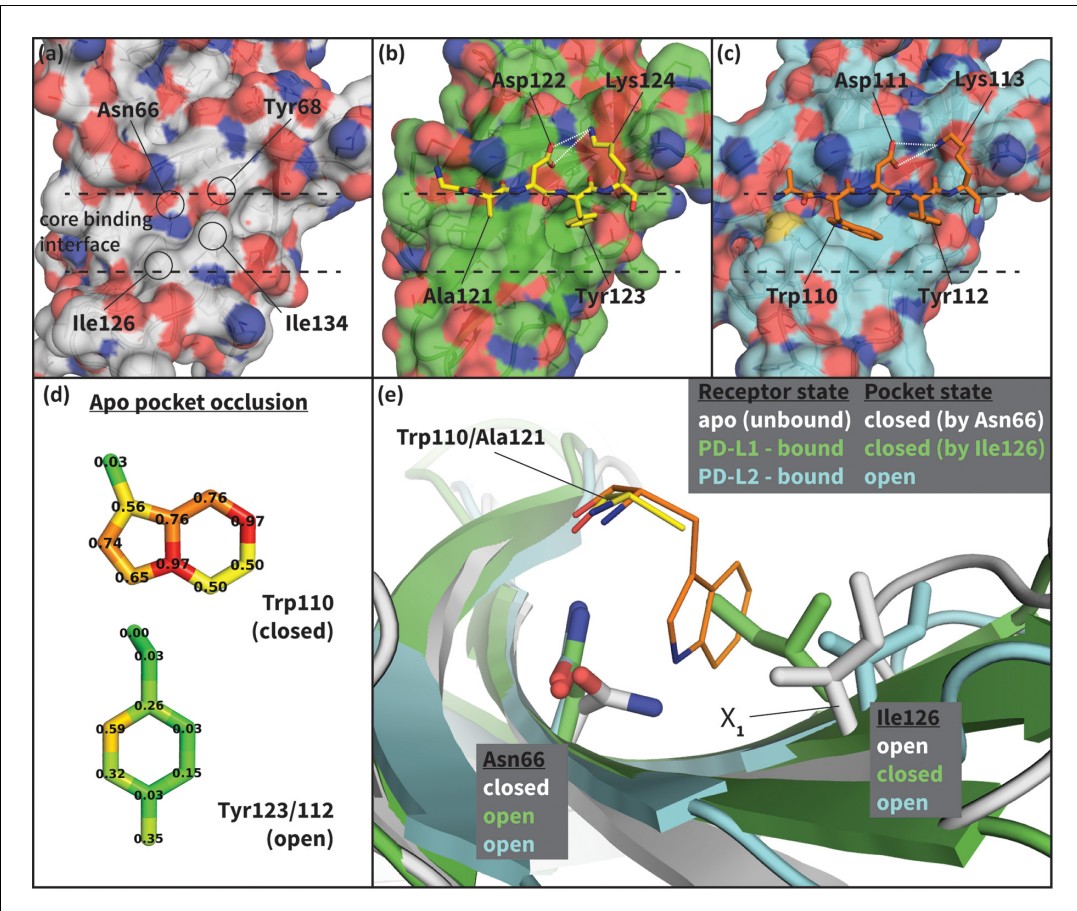

**Figure 2.** Flexibility of the PD-1 binding interface. (**a**) The apo PD-1 binding interface (*Cheng et al., 2013*), showing a flat, polar, core binding interface. Surface residues that shape the core binding interface are labelled. (**b**) The core PD-1 (green) - PD-L1 (yellow) binding interface, showing a flat hydrophobic receptor surface (*Zak et al., 2015*). White dashed lines indicate hydrogen bonds between PD-L1 side chains. (**c**) The core PD-1 (cyan) – PD-L2 (orange) binding interface, showing a large hydrophobic receptor cavity (*Lázár-Molnár et al., 2008*). White dashed lines indicate hydrogen bonds between PD-L2 side chains. Note that the conserved anchor residue Tyr123/112 is present in both (**b**) and (**c**). (**d**) Fractional occlusion of each bound-like Trp110 and Tyr123/112 atom position in the NMR ensemble of apo PD-1. Numerical values at each atom position denote the fraction of NMR frames that overlap, or 'occlude', that position (see Materials and methods for full details of how fractional occlusion is calculated). Aside from the $C_\beta$, the Trp110 pocket is mostly occluded in the apo PD-1 ensemble, whereas the Tyr123/112 anchor pocket is largely open. (**e**) Overlay of apo, PD-L1–bound, and PD-L2–bound structures of PD-1 defining the 'open' and 'closed' states of PD-1 residues Asn66 and Ile126 in relation to the open and closed states of the Trp110 binding pocket.

The following figure supplements are available for figure 2:

**Figure supplement 1.** The cognate ligands of PD-1.

**Figure supplement 2.** Modulation of PD-1's flexible interface cavity.

(*Cheng et al., 2007*). It is reasonable to assume that the flexibility of the Trp110 pocket, and the fact that in the apo state it is largely occluded by the unmatched, polar NH2 group of Asn66 (*Figure 2a,d*), would present significant obstacles to traditional structure-based drug-design methods attempting to target this cavity (*Cozzini et al., 2008*). Thus, efforts to model the binding mechanism of PD-1 would not only shed light on nature's design principles for flexible and promiscuous

protein-protein interfaces, but they may also offer novel avenues for pursuing rational drug design against this and other high-impact targets.

To study the mechanism of PD-1 binding, we use molecular dynamics simulations (MDs) to identify and quantify the effects of intermolecular interactions on the PD-1 binding interface. We first estimate $G_{BL}^{apo}$ for the free receptor and demonstrate that apo$_{BL}$ states are exceedingly rare. We then estimate $G_{BL}^{EC}$ for PD-1 interacting with various peptide constructs that mimic distinct subsets of ligand interface motifs (*Figure 3*) and identify the critical features that trigger shifts in the PD-1 conformational ensemble toward the bound-like states. By quantifying the energetic contribution of each triggering contact in the EC$_{NBL}$, we rationalize how PD-1 uses flexibility to simultaneously achieve both promiscuity, that is, binding to multiple ligands, and specificity. We show that a conserved set of three contacts in the PD-1 encounter complexes with PD-L1/2 progressively lowers the free energy of bound-like receptor states with respect to the non-bound-like state. These molecular triggers reshape the non-bound-like hydrophilic interface around Asn66 into a bound-like hydrophobic surface. A fourth contact that differs by a single atom stabilizes this surface into either a shallow patch that interacts with Ala121 in PD-L1, or a deep cavity that buries Trp110 in PD-L2.

We find that these triggers, which include the anchor Tyr123/112 in PD-L1/PD-L2 (*Figure 2b,c,d*) (*Rajamani et al., 2004*), are highly conserved across species (*Lázár-Molnár et al., 2008*) and drive quantitatively similar, kinetically efficient downhill binding pathways. The importance of these triggers is underscored by the PD-1 – targeting, anticancer antibody pembrolizumab, which evolved via a distinct evolutionary pathway yet, as we show, exploits some of the same triggering machinery as PD-1's natural ligands. Finally, we suggest how these induced-fit triggers could be used in rational, small-molecule drug discovery by studying the binding mode of a potent macrocyclic PD-1 inhibitor. Collectively, our findings demonstrate how nature exploits structural flexibility to achieve selective binding promiscuity in regulatory proteins.

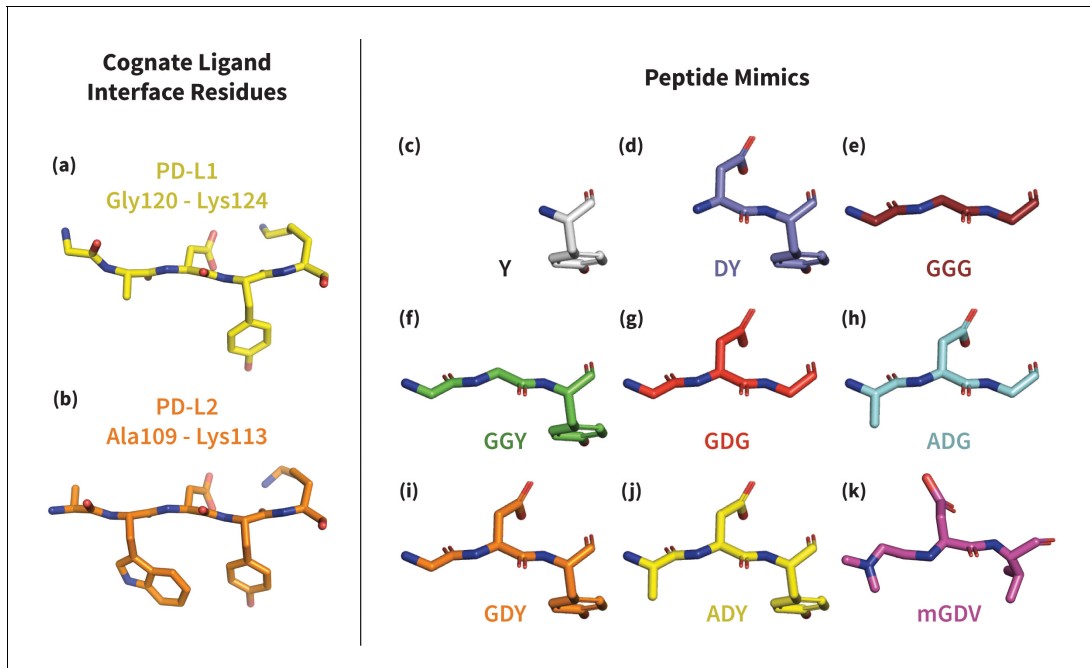

**Figure 3.** Structures of PD-L1/2 – mimicking peptides used to probe PD-1 interface dynamics. Left: core interface binding residues of (**a**) PD-L1 and (**b**) PD-L2 in their bound-like conformations. Right: peptides that were simulated in the presence of apo PD-1 in order to identify the triggers of induced fit interface deformations: (**c**) Y, (**d**) DY, (**e**) GGG, (**f**) GGY, (**g**) GDG, (**h**) ADG, (**i**) GDY, (**j**) ADY, and (**k**) mGDV.

## Results

### Open and closed states of PD-1 Asn66 and Ile126 describe a hydrophilic or hydrophobic interface

Analysis of aligned PD-1 structures (*Figure 2*) led us to classify the bound-like and non-bound-like conformational states using two binary order parameters defined by the 'open' or 'closed' states of Asn66 and Ile126. Namely, for a non-bound-like interface Asn66 is closed and Ile126 is open; for the PD-L1-specific bound-like state Asn66 is open and Ile126 is closed; and for the PD-L2-specific bound-like state both Asn66 and Ile126 are open (*Figure 2e*). In the PD-L1–bound state, the interface exhibits a large hydrophobic patch that interacts with the side chain of ligand interface residue Ala121 (*Figure 2b*). In the PD-L2–bound state, the interface exhibits a deep hydrophobic cavity that buries ligand residue Trp110 (*Figure 2c*). Neither this hydrophobic patch nor deep cavity is sampled in the apo PD-1 NMR ensemble, where, instead, the closed state of Asn66 blocks the Trp110-binding pocket by exposing its NH2 group (*Figure 2a,e*, *Figure 2—figure supplement 2*), making a hydrophilic site. MDs of apo PD-1 confirm that Asn66 remains closed (*Figure 4a*), stabilized by a hydrogen bond with Lys78 that is also present in NMR structures (*Figure 5a*). These findings suggest that specific interactions between apo PD-1 and a nearby ligand might be required to open Asn66 and reshape the hydrophilic interface into its hydrophobic bound-like states.

### Bound-like conformations of unbound Tyr123/112 in PD-L1/2 facilitates molecular recognition

For both induced fit and conformational selection, the association of the apo receptor and ligand is driven mainly by diffusion (*DeLisi, 1980*; *Northrup and Erickson, 1992*). It has been shown that often protein-protein interactions stabilize the initial encounter complex through the burial of a bound-like anchor motif on the ligand (*Rajamani et al., 2004*), which allows subsequent, longer timescale intermolecular interactions to take shape. Co-crystal structures, MDs and docking studies of PD-L1/2 suggest that the homologous interface residues Tyr123/112 (see *Figure 2b,c*) may serve as anchors. Specifically, MDs of apo PD-L1/2 show that Tyr123/112 remain within 0.5 Å heavy atom RMSD of their bound-like conformations 88 ± 16% of the time. Furthermore, the Tyr123/112-binding pocket is unobstructed in the apo PD-1 NMR ensemble (*Figure 2d*), facilitating immediate burial of the side chain upon association. Docking exercises also point to the stabilizing role of the Tyr anchor. Namely, ClusPro (*Comeau et al., 2004*) successfully re-docked the wild-type human PD-1 – PD-L1 co-crystal (*Zak et al., 2015*), but it failed for single-residue PD-L1 mutants Y123G and Y123A (*Table 1*). Collectively, these results suggest an anchor role for Tyr123/112 that facilitates molecular recognition between non-bound-like apo PD-1 and its ligands (as sketched in *Figure 1*).

### Conserved PD-L1/2 Asp122/111 form a specific intermolecular hydrogen bond network that opens PD-1 Asn66 and switches the receptor interface from hydrophilic to hydrophobic

Co-crystal structures of bound PD-1 exhibit an open Asn66 that forms two hydrogen bonds: the first with the backbone oxygen of homologous PD-L1/2 Ala121/Trp110 and the second with either PD-1 Tyr68 (human PD-1 - PD-L1 complex) or a crystal water (murine PD-1 – PD-L2 complex) (*Figure 5b, c*). MDs of PD-1 in complex with a GGG peptide positioned to mimic the backbone of PD-L1/2 residues ADY123 and WDY112, respectively, show that Asn66 fluctuates back and forth between a bound-like open state, where it makes the aforementioned backbone hydrogen bond to the GGG peptide, and the non-bound-like closed state, where it is bonded to PD-1 Lys78 (*Figure 4a*). On the other hand, simulations with a GDG peptide show that the Asp122/111 mimic forms a hydrogen bond to the Tyr68 OH group, stabilizing a Tyr68 rotamer that can simultaneously hydrogen bond to the NH2 of Asn66 (*Figure 5d*). Together, this Asn66 – Tyr68 hydrogen bond and the aforementioned Asn66 – backbone hydrogen bond stabilize the bound-like open state of Asn66 (*Figure 4a*).

The robust, four-membered hydrogen bond network between the Ala121/Trp110 backbone mimic, Asn66, Tyr68, and the Asp122/111 mimic that we observe in GDG MDs is fully consistent with all available structures and mutagenesis experiments. Namely, the hydrogen bonds rationalize the conservation of Asp122/111 in all known PD-L1/2 sequences and explain PD-L2 mutagenesis studies showing that the D111A mutation abolishes binding to PD-1 (*Lázár-Molnár et al., 2008*).

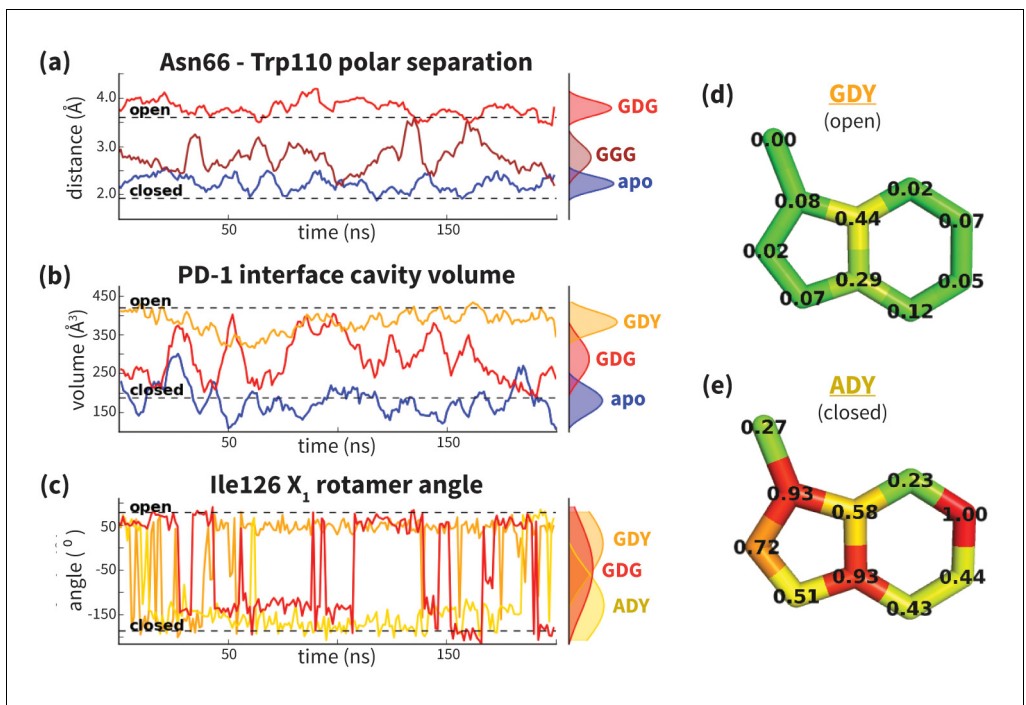

**Figure 4.** Dynamics of PD-1 binding interface in the presence of different ligands. (**a**) Rolling averages of distance between Trp110_NE1 (from bound PD-L2) and Asn66_ND2 from MDs of apo PD-1 (blue) alone and interacting with GGG (maroon) and GDG (red) peptides. Only GDG peptide sequesters Asn66 away from Trp110 binding pocket. (**b**) Rolling averages of PD-1 binding cavity volume from simulations of apo PD-1 alone (blue) and interacting with GDG (red) and GDY (orange) peptides shows that only GDY stabilizes an open cavity. (**c**) Ile126 $X_1$ rotamer angle from MDs of apo PD-1 interacting with GDG (red), GDY (orange), and ADY (yellow) peptides. Peptide ADY and GDY position Ile126 in the closed and open states, respectively (as in *Figure 2e*). Replicate trajectories for panels a, b, and c are shown in *Figure 4—figure supplement 2*. (**d**) Fractional occlusion of each bound-like Trp110 atom position in simulations of PD-1 interacting with the GDY peptide show an open Trp110-binding pocket. The fractional occlusion of a Trp110 atom position is defined as the percentage of simulation frames in which a PD-1 atom overlaps, or 'occludes', that position (see Materials and methods for full details of how fractional occlusion is calculated). (**e**) Fractional occlusion of each bound-like Trp110 atom position in simulations of PD-1 interacting with the ADY peptide show a closed Trp110-binding pocket.

The following source data and figure supplements are available for figure 4:

**Source data 1.** Excel workbook containing all the simulation trajectory data plotted in *Figure 4*, *Figure 4—figure supplement 2*, and *Figure 4—figure supplement 3*.

**Figure supplement 1.** Apo PD-1 interactions with GDY peptide opens a hydrophobic cavity.

**Figure supplement 2.** Replicate trajectories from *Figure 4a,b,c*.

**Figure supplement 3.** Dynamics of PD-1 binding cavity in the presence of different anchor substitutes.

MDs of apo PD-L1/2 further support the importance of Asp122/111 interactions in the encounter complex by showing that this side chain remains within 0.4 Å RMSD of its bound-like conformation 82 ± 25% of the time. The stabilization of the bound-like Asp122/111 side chain in simulation is achieved via hydrogen bonds with the neighboring Lys124/113, bonds which are also observed in bound cocrystal structures of PD-1 (*Figure 2b,c*). The importance of this stabilizing interaction is underscored by the fact that the K124S and K113A point mutations in PD-L1 and PD-L2, respectively, both abolish binding to PD-1 (*Lázár-Molnár et al., 2008*; *Lin et al., 2008*).

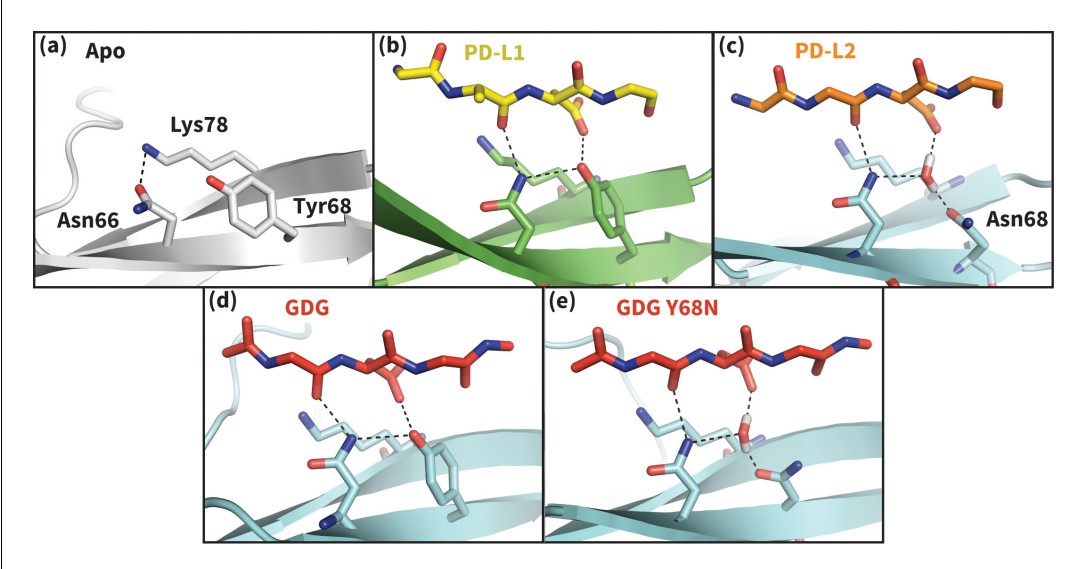

**Figure 5.** Hydrogen bond network of PD-1 Asn66 in different contexts. (**a**) NMR structure of the dominant apo, non-bound-like state of the human PD-1 interface (*Cheng et al., 2013*). Asn66 is in the closed state, forming a single hydrogen bond with Lys78. (**b**) Cocrystal structure of the human PD-1 – PD-L1 complex (*Zak et al., 2015*). PD-1 bound-like interface shows Asn66 in the open state, forming two hydrogen bonds with the ligand Ala121 backbone and the neighboring Tyr68. For clarity, only relevant ligand atoms are shown. (**c**) Cocrystal structure of the murine PD-1–PD-L2 complex (*Lázár-Molnár et al., 2008*). PD-1 bound-like interface shows Asn66 is in the open state, forming two hydrogen bonds with the ligand Trp110 backbone and a crystal water stabilized by neighboring residue Asn68. (**d**) Simulation snapshot of human PD-1 interacting with the GDG peptide, showing the same hydrogen bond network as in (**b**). (**e**) Simulation snapshot of human PD-1 Y68N mutant interacting with the GDG peptide, showing the same water-mediated hydrogen bond network as in (**c**).

PD-1 ligands open Asn66 by offering two novel hydrogen bonds (with the Ala121/Trp110 back-bone and Tyr68) that out-compete the single Lys78 hydrogen bond that stabilizes the closed state. Interestingly, the one known PD-1 sequence that diverges at the Tyr68 position is murine PD-1, which has a Y68N mutation. The murine PD-1–PD-L2 co-crystal shows that although the shorter Asn68 side chain cannot hydrogen bond directly to Asn66 or Asp111, it hydrogen bonds to a crystal

**Table 1.** Anchor Tyr123 is key determinant of bound-like docked conformations. Backbone RMSD of top 10 ClusPro (*Comeau et al., 2004*) predicted PD-L1 binding modes to the human PD-1–PD-L1 cocrystal (PDB: 4ZQK). RMSDs shown for docked wild type human PD-L1 (WT) and for docked PD-L1 anchor mutants Y123G and Y123A.

| ClusPro model | Docked PD-L1 backbone RMSD (Å) to 4ZQK PD-L1 | | |
| --- | --- | --- | --- |
| | WT | Y123G | Y123A |
| 0 | **4.65** | 8.8 | 49.7 |
| 1 | 54.0 | 38.2 | 49.1 |
| 2 | 49.5 | 49.1 | 39.2 |
| 3 | 47.5 | 40.4 | 40.4 |
| 4 | 39.4 | 49.4 | 48.5 |
| 5 | 48.0 | 40.07 | 53.2 |
| 6 | 45.8 | 53.2 | 49.5 |
| 7 | 40.6 | 46.5 | 48.1 |
| 8 | 48.6 | 47.8 | 47.6 |
| 9 | 50.7 | 48.7 | 50.4 |

water molecule that forms the same hydrogen bond network as Tyr68 (*Figure 5c*). MDs of a human Y68N PD-1 mutant and the GDG peptide suggest a functional equivalence of Asn68 to Tyr68: the Asn68 side chain spontaneously recruits a stable water to the co-crystal position that then opens Asn66 via a specific hydrogen bond network analogous to that formed by Tyr68 (*Figure 5e*).

## ADY/GDY ligand motifs stabilize distinct bound-like states for PD-L1/2

While GDG MDs show an open Asn66 (*Figure 4a*) that exposes a hydrophobic surface, this surface remains flexible and fluctuates between a deep open cavity and closed shallow patch (*Figure 4b*). Contrary to the GGG MDs that exhibited open-closed fluctuations of Asn66 (*Figure 4a*), the pocket instability observed in GDG MDs is caused by open-closed fluctuations of PD-1 residue Ile126 (*Figure 4c*). In contrast, MDs show that the GDY peptide stabilizes the open states of both Asn66 and Ile126 and maintains the open hydrophobic interface cavity seen in the PD-L2 bound-like state of PD-1 (*Figure 4b,c,d*). Comparison of the GDG and GDY MDs reveal that the Tyr side chain serves as a 'wedge' that stabilizes the flexible loop surrounding Ile134 into a bound-like configuration that is observed in both the PD-L1 and PD-L2 co-crystal structures (*Figure 6*). In the presence of the GDY peptide, the bound-like Ile134 makes a hydrophobic contact with the long arm of Ile126, which pulls the latter residue out of the pocket and stabilizes its open state (*Figure 4—figure supplement 1*).

Although the PD-L1 interface exhibits the GDY scaffold, Ile126 is closed in the PD-L1-specific $EC_{BL}$ state, suggesting that an additional ligand motif not contained in the GDY scaffold is responsible for closing the pocket. MDs with an ADY peptide that mimics Ala121 show that the extra $C_\beta$ carbon of the Ala side chain out-competes Ile134 for the long arm of Ile126, stabilizing its closed state (*Figure 4c,e*). Interestingly, MDs with GDG and ADG peptides both show similarly unstable open-closed fluctuation of Ile126 (see *Figure 7* below), which suggests that the effect of the Ala121 $C_\beta$

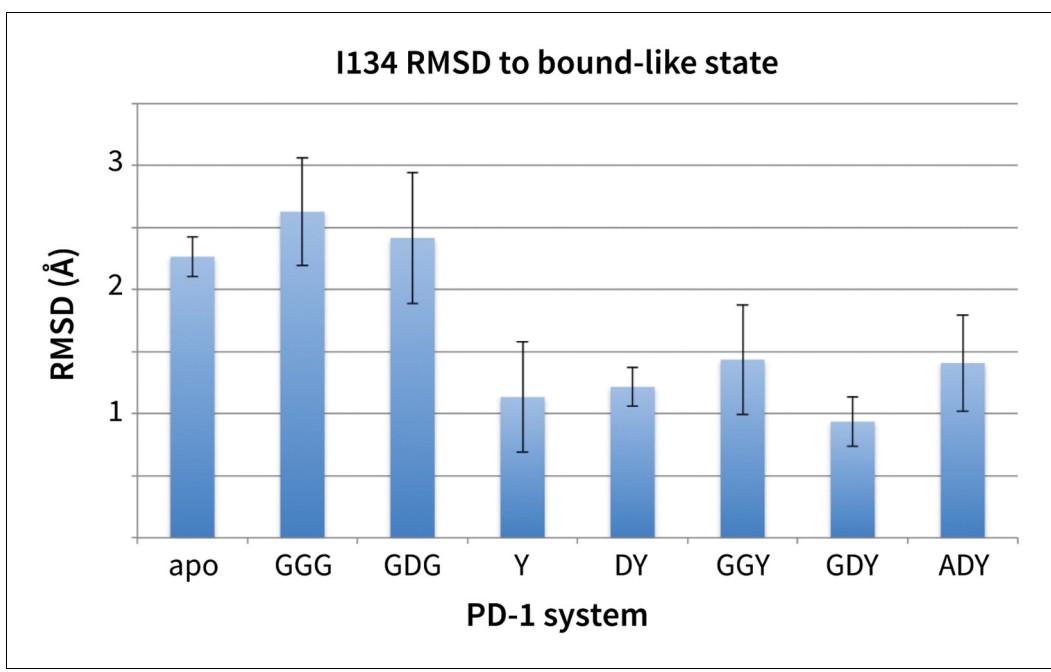

**Figure 6.** Stabilization of bound-like Ile134 by conserved tyrosine (**Y**) anchor. Average and standard deviation heavy atom RMSD of PD-1 Ile134 to the PD-L1/2 bound-like state (measured from human PD-1 – PD-L1 cocrystal, 4ZQK; Ile134 has <0.2 Å heavy atom RMSD between 4ZQK and the PD-L2 cocrystal 3BP5). Data are shown for three 200ns replicate simulations for each system, including apo human PD-1 and PD-1 interacting with various peptides.

The following source data is available for figure 6:

**Source data 1.** Excel workbook with a single sheet containing the numerical RMSD data shown in the *Figure 6* bar chart.

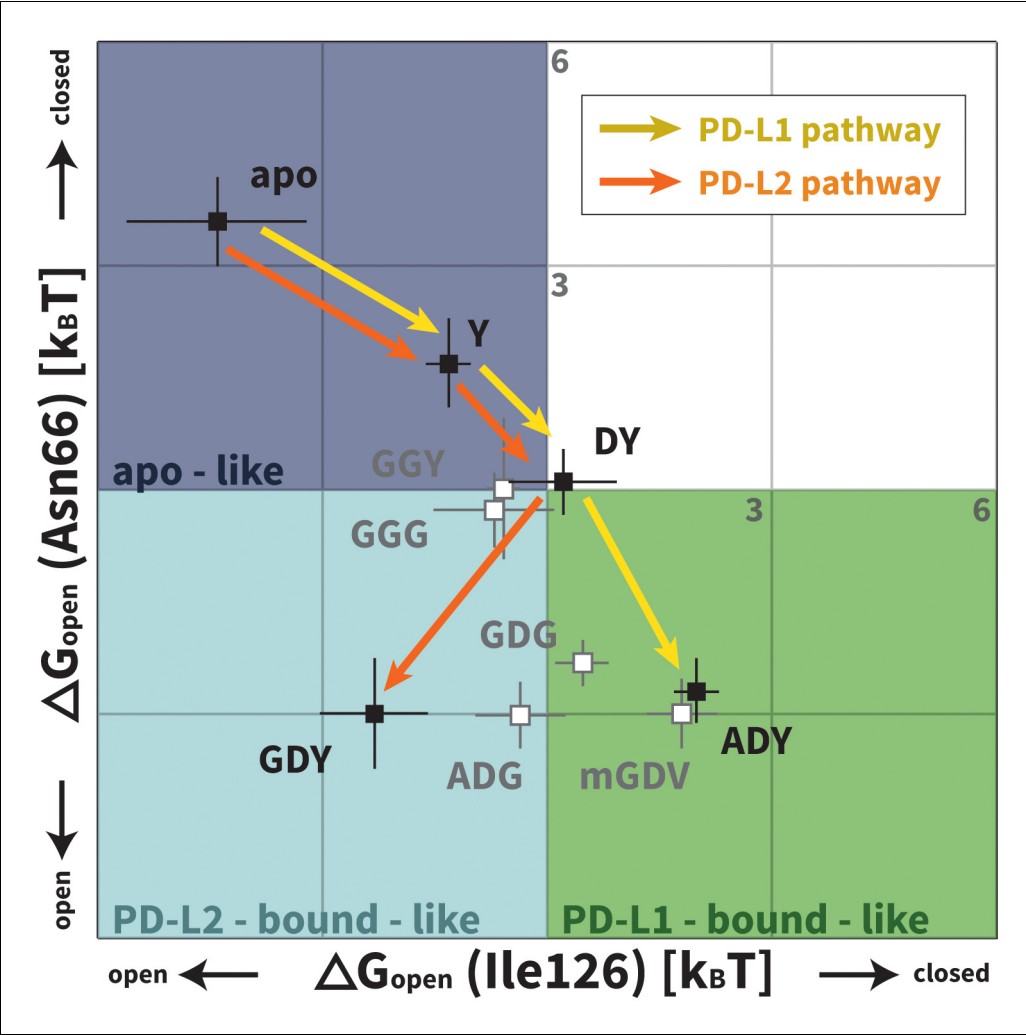

**Figure 7.** Downhill binding pathways of PD-1 triggers of induced fit for each cognate ligand. Points on the plot represent average and standard deviation equilibrium free energy differences (from three replicate simulations) between the open and closed states of receptor residues Asn66 and Ile126 for apo PD-1 and PD-1 interacting with nine distinct ligand-mimicking peptides. The corresponding numerical values can be found in *Table 2*. Yellow and orange lines represent the ligand-specific induced fit binding pathways from the apo receptor ensemble to the PD-L1 and PD-L2 bound-like ensembles, respectively.
The following source data is available for figure 7:

**Source data 1.** Excel workbook with a single sheet containing the numerical $\Delta G_{open}$ data plotted in *Figure 7*.

carbon on Ile126 dynamics only emerges in the presence of the anchor Tyr123/112. Thus, in addition to facilitating molecular recognition, stabilization of the Ile134 loop by the burial of Tyr123/112 is shown to enable ligand-specific induced fit responses by the PD-1 interface.

## PD-L1/2 triggers ADY/GDY produce energetically downhill induced fit binding pathways

We applied Maxwell-Boltzmann statistics to our peptide simulations (see Materials and methods) to quantify the role played by each trigger in the structural transitions at the PD-1 interface. We evaluate $\Delta G_{open}$, that is, the free energy differences between the open and closed states of Asn66/Ile126 for PD-1 in isolation and PD-1 interacting with nine different peptides representing distinct PD-L1/2 interface motifs (*Figure 3*, *Table 2*; note that $\Delta G_{open}$ and $\Delta G_{BL}$ are trivially related). These $\Delta G_{open}$

**Table 2.** Free energy difference between the non-bound-like and bound-like states of PD-1 interface residues Asn66 and Ile126 in various systems. Listed values show the average and standard deviation of $G_{BL}$ (from three replicate simulations) for Asn66 and Ile126 in the different PD-1 systems. Since the bound-like state of Ile126 is closed when PD-L1 – bound and open when PD-L2 - bound, the $\Delta G_{BL}$ values for this residue take opposite signs. The trivial relationship between $\Delta G_{BL}$ and $\Delta G_{open}$ are indicated for each column. Values shown are in units of $k_B$T, with T = 300 K.

| PD-1 Simulation | Pd-l1 / PD-L2 $\Delta \mathbf{G_{BL}}(\mathbf{Asn}66)$ $\Delta G_{open}(Asn66)$ ($k_B$T) | PD-L1 $\Delta G_{\mathbf{BL}}(Ile126)$ $-\Delta G_{open}(Ile126)$ ($k_B$T) | PD-L2 $\Delta G_{\mathbf{BL}}(Ile126)$ $\Delta G_{open}(Ile126)$ ($k_B$T) |
|---|---|---|---|
| apo ($\Delta G_{BL}^{apo}$) | 3.6 ± 0.60 | 4.4 ± 1.2 | −4.4 ± 1.2 |
| Y ($\Delta G_{BL}^{Y}$) | 1.7 ± 0.61 | 1.3 ± 0.3 | −1.3 ± 0.3 |
| DY ($\Delta G_{BL}^{DY}$) | 0.11 ± 0.44 | −0.21 ± 0.74 | 0.21 ± 0.74 |
| GGG ($\Delta G_{BL}^{GGG}$) | −0.28 ± 0.52 | 0.7 ± 0.8 | −0.7 ± 0.8 |
| GGY ($\Delta G_{BL}^{GGY}$) | 0.02 ± 0.17 | 0.58 ± 0.22 | −0.58 ± 0.22 |
| GDG ($\Delta G_{BL}^{GDG}$) | −2.3 ± 0.32 | −0.46 ± 0.36 | 0.46 ± 0.36 |
| ADG ($\Delta G_{BL}^{ADG}$) | −3.0 ± 0.45 | 0.35 ± 0.60 | −0.35 ± 0.60 |
| GDY ($\Delta G_{BL}^{GDY}$) | −3.0 ± 0.7 | 2.3 ± 0.72 | −2.3 ± 0.72 |
| ADY ($\Delta G_{BL}^{ADY}$) | −2.7 ± 0.44 | −2.0 ± 0.36 | 2.0 ± 0.36 |
| mGDV ($\Delta G_{BL}^{mGDV}$) | −3.0 ± 0.47 | −1.8 ± 0.48 | 1.8 ± 0.48 |

values are plotted in *Figure 7*. Remarkably, the ADY and GDY motifs, respectively, shift the ratio of our predefined bound-like to non-bound-like states from 1: 44 ± 24 (based on $\Delta G_{open}^{apo}(Asn66)$) to 7.4 ± 2.8: one for the PD-L1 bound-like state (based on $\Delta G_{open}^{ADY}(Ile126)$) and 12 ± 9.6: one for the PD-L2 bound-like state (based on $\Delta G_{open}^{GDY}(Ile126)$). More importantly, we show that each triggering contact monotonically lowers the relative free energy of ligand-specific bound-like states starting from no contacts (apo), to the first, conserved contact with the anchor (Y), to the second, conserved contact with Asp122/111 (DY), to the final, unconserved contact with the backbone O of A/G in the complete triggering motifs (ADY/GDY) (*Figure 7*). The fact that these downhill binding pathways do not encounter energy barriers strongly suggests that the PD-1 binding mechanism is primarily one of induced fit (see *Figure 1*).

In the apo simulation Asn66 is closed ($\Delta G_{open}^{apo}(Asn66) \approx 3.6\ k_BT$), repelling Ile126 into an open conformation ($\Delta G_{open}^{apo}(Ile126) \approx -4.4\ k_BT$). Docking of the Tyr anchor (Y) and formation of the encounter complex destabilizes the non-bound-like apo PD-1 interface, causing increased open-closed fluctuations in both Asn66 and Ile126. The subsequent docking of Asp122/111 (DY) allows Tyr68 to compete with Lys78 to form one hydrogen bond with Asn66, causing it to swap back and forth between open and closed ($\Delta G_{open}^{DY}(Asn66) \approx 0$). Fluctuations of Asn66 correlate with simultaneous fluctuations of Ile126 ($\Delta G_{open}^{DY}(Ile126) \approx 0$). Adding the adjacent Ala121/Trp110 backbone from PD-L1/2 (ADY/ GDY) provides the second hydrogen bond for the NH2 of Asn66 that fully stabilizes its open state ($\Delta G_{open}^{GDY/ADY}(Asn66) \approx -3.0\ k_BT$). With Asn66 open, the Cβ atom of Ala121 modulates Ile126 dynamics. When present (ADY), the $C_\beta$ hydrophobically recruits Ile126 into the closed pocket state ($\Delta G_{open}^{ADY}(Ile126) \approx 2.0\ k_BT$). Without $C_\beta$ (GDY), Ile126 remains open ($\Delta G_{open}^{GDY}(Ile126) \approx -2.3\ k_BT$).

Our $\Delta G_{open}$ calculations also quantify the critical role of the anchor residue Tyr123/112 in ensuring the ligand specificity of PD-1 interface deformations. This is demonstrated by the fact that GDY and ADY peptides impose clear differential influence on the dominant rotamer state of Ile126, while for both GDG and ADG, Ile126 fluctuates about evenly between the open and closed state ($\Delta G_{open}^{GDG/ADG}(Ile126) \approx 0$) (*Figure 7*).

## Encounter complex simulations suggest chronology of induced fit triggering interactions

We ran MDs of the PD-L1/2 encounter complexes starting from docked poses of apo PD-1 and the interacting domains of PD-L1/2 that anchored Tyr123/Y112 (see Supporting Materials and methods). Encounter complex MDs recapitulated the triggering mechanisms we identified in our peptide simulations and their resulting PD-1 interface transitions from the $EC_{NBL}$ to the ligand-specific $EC_{BL}$ states. The chronology for these interactions (*Table 3*) is the same for both ligands. Consistently, the first interaction to take place after docking the conserved anchor is the formation of the hydrogen bond between receptor residue Tyr68 and ligand residue Asp122/111. This is followed by stabilization of Asn66 in the open pocket state via hydrogen bonds with neighboring Tyr68 and the ligand Ala121/Trp110 backbone. The Ala121/Trp110 side chains then proceed to stabilize a closed/open hydrophobic pocket. Note that the Trp in the WDY motif of PD-L2 readily fills the hydrophobic pocket as the XDY motif latches and opens Asn66 (*Figure 8*). Consistent with a downhill free energy induced fit mechanism, the realization of these four contacts takes less than 10 ns total. On a longer timescale, encounter complex simulations demonstrate the formation of secondary hydrogen bonds at the interface periphery that are also observed in co-crystal structures of human and murine PD-1. These secondary hydrogen bonds, including the bond from PD-1 Lys78 to PD-L1/2 Phe19/21 and from Gln75 to Arg125/Tyr114 (*Figure 9*), were consistently observed to form approximately 10 nanoseconds after the aforementioned Asn66 and Tyr68 hydrogen bonds (*Table 3*), suggesting that $EC_{BL}$ contacts shaped by the triggers of induced fit are enough to drive the subsequent transition to the HAC.

## PD-1 – targeting antibody validates the critical role of Asn66 and suggests an anchor-independent binding mechanism with closed Ile126 and Ile134

Recently, two FDA-approved PD-1–targeting antibodies have emerged as part of a new generation of anticancer immune checkpoint inhibitors. Published crystal structures of one of these antibodies, pembrolizumab, bound to extracellular PD-1 show a hydrophobic receptor binding surface that overlaps that which binds PD-L1/2 (*Figure 10b*) (*Horita et al., 2016*; *Lee et al., 2016*; *Na et al., 2017*). Comparison of the pembrolizumab – PD-1 interface to the PD-L1 – PD-1 interface using the Fast-Contact server (*Champ and Camacho, 2007*) highlights several differences in the main contacts that characterize the two binding modes (*Figure 10a*, *Tables 4* and *5*). Remarkably, the pembrolizumab-bound crystal structures reveal that the antibody stabilizes the same open state of Asn66 as PD-L1/2 using an analogous hydrogen bond network (*Figure 10c*). The fact that this antibody, designed via a distinct evolutionary pathway, shares PD-L1/2's mechanism for opening Asn66 and revealing a hydrophobic binding surface (*Figure 2a,b,c*, *Figure 10b*) underscores the role of this specific interaction in PD-1 interface remodeling.

Although pembrolizumab's interaction with Asn66 mimics the native-like contacts of PD-L1/2, the antibody-bound receptor exhibits a novel configuration of Ile134, with both Ile126 and Ile134 in inward-flipped, 'closed' states (*Figure 10b*). The result is a large hydrophobic surface where, like in

---

**Table 3.** Chronology of the formation of intermolecular interactions between PD-1 and PD-L1/2 in encounter complex simulations. Listed values show the average and standard deviation time to formation (from three replicate simulations) of various inter- and intramolecular hydrogen bonds following the burial of the ligand anchor and formation of the key Tyr68–Asp122/111 hydrogen bond.

| Hydrogen bond | $\Delta t$ (ns) after Tyr68 – Asp122/111 hydrogen bond formation | |
| | PD-1 - PD-L1 Encounter Complex | PD-1 - PD-L2 Encounter Complex |
| --- | --- | --- |
| Asn66 – Ala121/Trp110 | 6.3 ± 2.9 | 6.7 ± 7.2 |
| Asn66 – Tyr68 | 5.0 ± 1.7 | 8.3 ± 7.5 |
| Gln75 – Arg125/Tyr114 | 15 ± 7.8 | 17 ± 11 |
| Lys79 – Phe19/21 | 13 ± 15 | 15 ± 20 |

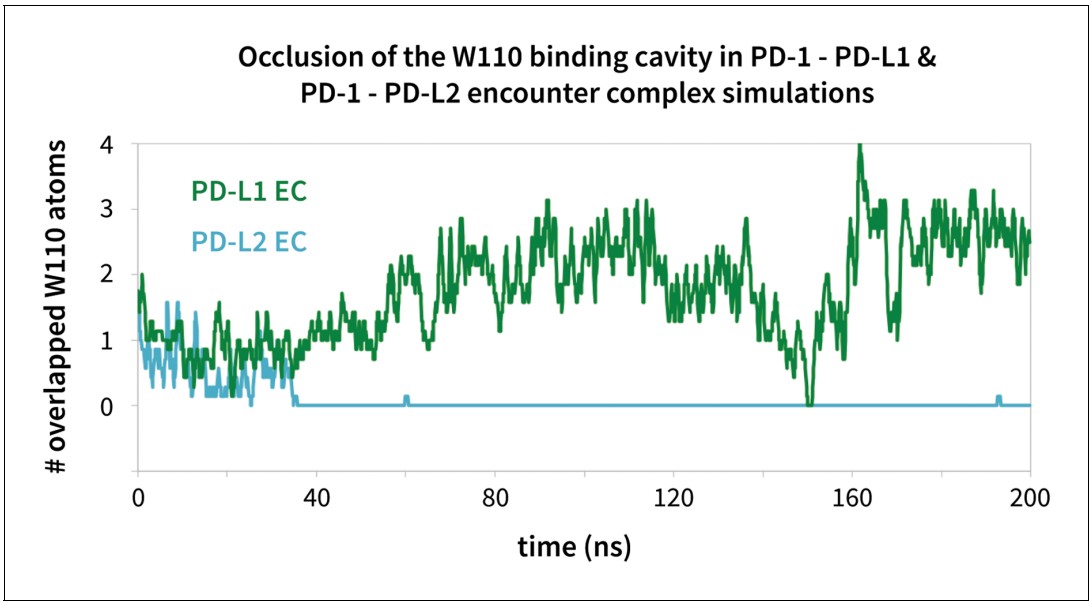

**Figure 8.** Modulation of the PD-1 interface binding cavity in encounter complex simulations with PD-L1 and PD-L2. Plot shoes the (rolling average) number of atoms in the bound-like Trp110 side chain reference that are occluded by the PD-1 interface throughout encounter complex simulations with PD-L1/2 (see Materials and methods for full details of how occlusion is calculated). Both encounter complexes begin with a closed Trp110 pocket, as this is the dominant state of apo PD-1. The PD-L2 trigger then stabilizes the hydrophobic cavity (no overlap), while the PD-L1 trigger stabilizes the hydrophobic patch (significant overlap).

The following source data is available for figure 8:

**Source data 1.** Excel workbook with a single sheet containing the time-series Trp110 atom overlap data from the encounter complex simulations plotted in *Figure 8*.

the PD-L1–bound state, the closed Ile126 occludes the Trp110-binding pocket, but where, unlike the PD-L1/2–bound states, a closed Ile134 partially fills the Tyr/123/112 anchor cavity. In fact, pembrolizumab has no anchor analog. Instead, the Arg102 side chain extends along the PD-1 interface such

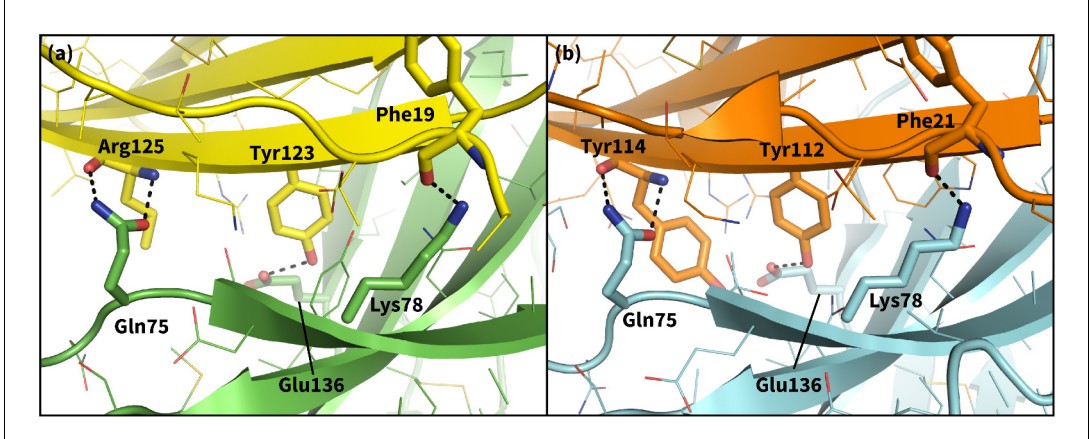

**Figure 9.** Secondary, non-triggering contacts in PD-1 encounter complexes. Specific hydrogen bonds observed in the PD-1 – PD-L1 (a) (*Zak et al., 2015*) and PD-1 – PD-L2 (b) (*Lázár-Molnár et al., 2008*) cocrystal structures. In simulation, these contacts form approximately 10 ns after triggering interactions and their resulting induced fit deformations of the receptor (*Table 3*). Note also that the conserved Tyr123/112 anchor forms identical hydrogen bonds with Glu136 in the PD-L1– and PD-L2–bound states.

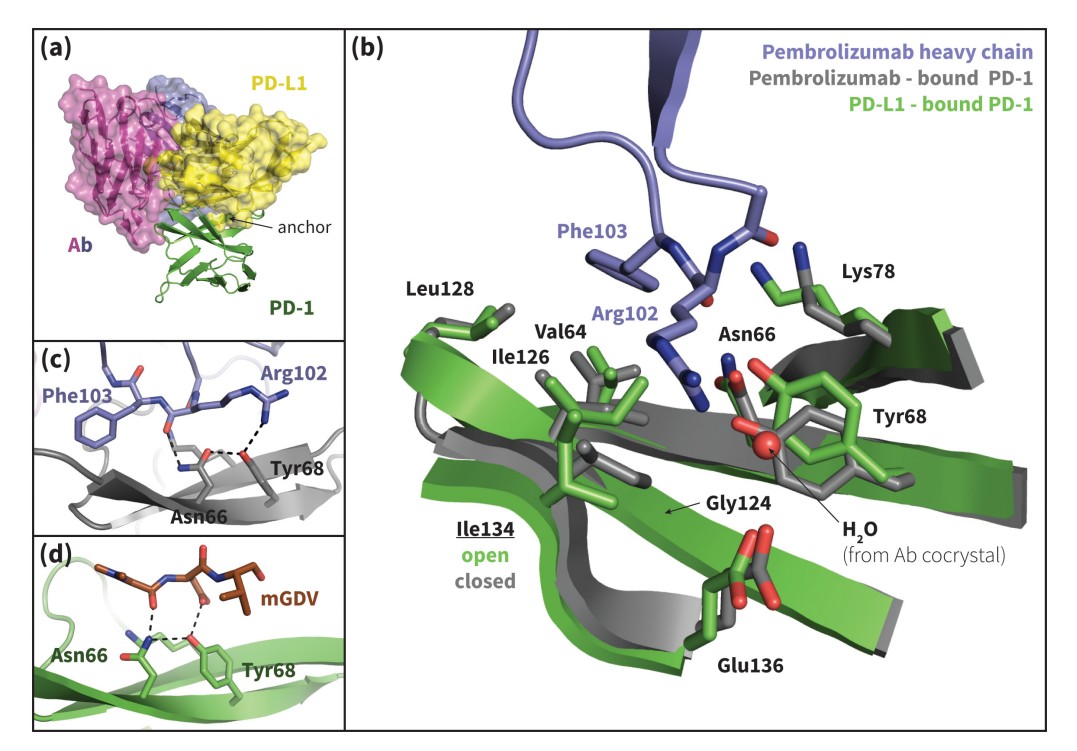

**Figure 10.** Pembrolizumab–bound PD-1 interface resembles PD-L1–bound interface with a closed Ile134. (**a**) Alignment of crystal structures of the pembrolizumab antibody (Ab) (*Horita et al., 2016*) and PD-L1 (*Zak et al., 2015*) binding modes, showing distinct but partially overlapping binding interfaces on PD-1. The light chain of the Ab is shown in magenta and the heavy chain is shown in purple. (**b**) Detailed comparison of the aligned Ab–bound (grey) and PD-L1–bound (green) PD-1 interfaces. Most receptor interface residues exhibit near-identical conformations, except Ile134 which is open when bound to PD-L1 but closed when bound to pembrolizumab. Heavy chain Ab interface residues are shown in purple. (**c**) Detail of the Ab–PD-1 interface, highlighting the hydrogen bond (hydrogen bond) network that stabilizes the open state of Asn66. This hydrogen bond network is functionally analogous to those observed in the PD-L1 and PD-L2–bound cocrystals (*Figure 5*), although the OD1 and ND2 atoms of Asn66 are flipped. (**d**) Simulation snapshot of human PD-1 interacting with the mGDV motif from Bristol-Myers Squibb macrocyclic PD-1 inhibitor, highlighting the canonical hydrogen bond network that opens Asn66.

The following figure supplements are available for figure 10:

**Figure supplement 1.** Model of potent Brystol-Myers-Squibb macrocyclic PD-1 inhibitor.

**Figure supplement 2.** Predicted interactions of Brystol-Myers-Squibb macrocyclic PD-1 inhibitor.

that the $C_Z$ carbon overlaps the $C_\gamma$ position of Tyr123/112 (*Figure 10—figure supplement 1*), and the NH1/2 groups hydrogen bond to a crystal water above the receptor interface (*Figure 10b*). In this configuration, the hydrophobic carbon chain of Arg102 forms a 'cap' above the closed Ile126 and Ile134, desolvating their hydrophobic interactions with each other and the neighboring Gly124 and stabilizing a flat hydrophobic surface (*Figure 10b*).

A similar closed conformation of Ile134 is observed in our MDs of PD-1 interacting with the GDG peptide (*Figure 4—figure supplement 1*). This is unsurprising: like pembrolizumab, the GDG peptide has the necessary machinery to trigger the opening of Asn66, but lacks an anchor 'wedge' that prevents the resulting inward collapse of Ile134. Results of the GDG MDs thus rationalize the pembrolizumab binding mode and suggest an anchor-independent induced fit PD-1 binding pathway: one in which the antibody opens Asn66 using the canonical hydrogen bond network and stabilizes the resulting flat hydrophobic interface by 'capping' the closed states of Ile126/134 with the carbon chain of Arg102.

**Table 4.** Top 5 PD-1 residues contributing to electrostatic energy when binding to PD-L1 and pembrolizumab. Binding energies were calculated using the FastContact web server (**Champ and Camacho, 2007**) and cocrystal structures of PD-1 bound to PD-L1 (**Zak et al., 2015**) and pembrolizumab (**Horita et al., 2016**).

| PD-L1–bound | | Pembrolizumab–bound | |
|---|---|---|---|
| Residue | Energy (kcal/mol) | Residue | Energy (kcal/mol) |
| Glu136[*] | −11.531 | Asp85[‡] | −8.367 |
| Asp77 | −5.073 | Ser87 | −3.629 |
| Lys78[†] | −4.266 | Asp77 | −2.417 |
| Gln75 | −4.027 | Tyr68 | −2.156 |
| Glu84 | −3.119 | Glu136 | −2.096 |

[*]The E136A mutation abolishes binding of PD-1 to PD-L1 and greatly reduces binding to PD-L2 (**Lázár-Molnár et al., 2008**).

[†]The K78A mutation abolishes binding of PD-1 to PD-L1 and greatly reduces binding to PD-L2 (**Lázár-Molnár et al., 2008**).

[‡]The D85G mutation abolishes binding of PD-1 to pembrolizumab (**Na et al., 2017**).

## Can molecular triggers be exploited to drug PD-1?

Although two PD-1-targeting antibodies already exist on the market, there are no small-molecule PD-1 inhibitors in clinical trial, despite the enormous interest in this blockbuster immunotherapy target (**Dömling and Holak, 2014**; **Couzin-Frankel, 2013**; **Zarganes-Tzitzikas et al., 2016**). Given that ligand-binding sites tend to be concave (**Laskowski et al., 1996**; **Liang et al., 1998**) and largely hydrophobic (**Cheng et al., 2007**), the undruggability of PD-1 might be due to the closed Asn66 and the resulting flat polar interface in the apo form (**Figure 2a**). However, the highly specific hydrogen bond network presented by PD-L1/2 and pembrolizumab strongly suggests a path to open Asn66 and transform the hard to drug hydrophilic patch into a hydrophobic one. Interestingly, Brystol-Myers-Squibb recently patented a 1.03 nM macrocyclic inhibitor of the PD-1–PD-L1 interaction (**Miller et al., 2014**). Although no mechanism of action has been described, the macrocycle includes a peptidic mGDV motif that is structurally similar to the aforementioned ADY induced fit trigger, with an N-methylated Gly and an Asp side chain that resemble PD-L1's Ala121 and Asp122, respectively (**Figure 10—figure supplements 1** and **2**). This alignment puts the mGDV motif's short Val

**Table 5.** Top 5 PD-1 residues contributing to desolvation energy when binding to PD-L1 and pembrolizumab. Binding energies were calculated using the FastContact web server (**Champ and Camacho, 2007**) and cocrystal structures of PD-1 bound to PD-L1 (**Zak et al., 2015**) and pembrolizumab (**Horita et al., 2016**).

| PD-L1–bound | | Pembrolizumab–bound | |
|---|---|---|---|
| Residue | Energy (kcal/mol) | Residue | Energy (kcal/mol) |
| Ile126[*] | −1.853 | Leu128[†] | −2.886 |
| Leu128[†] | −1.673 | Pro89 | −2.486 |
| Ile134[‡] | −1.361 | Val64 | −1.721 |
| Val64 | −0.463 | Pro130 | −1.586 |
| Ala132 | −0.37 | Pro83 | −1.131 |

[*]The I126A mutation greatly reduces binding of PD-1 to both PD-L1 and PD-L2 (**Lázár-Molnár et al., 2008**).

[†]The L128A mutation abolishes binding of PD-1 to PD-L1 and partially reduces binding to PD-L2 (**Lázár-Molnár et al., 2008**).

[‡]The I134A mutation abolishes binding of PD-1 t oPD-L1 and greatly reduces binding to PD-L2 (**Lázár-Molnár et al., 2008**).

side chain at the position of the much longer Tyr123 anchor, where it aligns with the $C_\Delta$ side chain carbon of pembrolizumab residue Arg112 (*Figure 10—figure supplement 1*).

Given the resemblance of the mGDV motif to the interface residues of both PD-L1 and pembrolizumab, we used our MDs method to evaluate whether this motif was capable of remodeling the apo, non-bound-like PD-1 interface into a bound-like state. We observed that mGDV opened Asn66 using a native-like hydrogen bond network analogous to those seen in previous simulations (*Figures 5*, *7* and *10d*). However, Ile126 and Ile134 dynamics mirrored those seen in the pembrolizumab cocrystal, with both sidechains favoring inward-flipped 'closed' configurations (*Figure 11*). Simulation trajectories showed that the short Val side chain of the mGDV motif, unlike the cognate Tyr123/112 anchors, did not penetrate deep enough into the PD-1 interface to be a 'wedge' stabilizing an open Ile134. Instead, like the carbon chain of pembrolizumab residue Arg102, the Val 'capped' stable hydrophobic interactions between a closed Ile134, a closed Ile126, and the neighboring Gly124.

Our GDG, ADG, GDY and ADY simulations demonstrated that precise regulation of the closed/open states Ile126 via the Ala121 $C_\beta$ is realized only when the Tyr123/112 anchor is buried (*Figure 7*). Thus, given that mGDV lacks an anchor, a natural question to ask is whether a Ile126 would be opened by a GDV peptide without the N-methyl group. Interestingly, MDs of PD-1 interacting with a GDV peptide revealed identical Ile126 and Ile134 dynamics to mGDV simulations (*Figure 11*), indicating that the N-methyl group was not recruiting Ile126 into the closed state in the style of Ala121 $C_\beta$. These results help to further illuminate the role of the conserved anchor Tyr123/112, which in its absence does not wedge Ile134 into the open state, disabling the capability of PD-1 to stabilize an open Ile126 and form a hydrophobic cavity at that site.

Compared to GDG simulations in which Ile126 fluctuated between open and closed (*Figure 4b, c*), in GDV simulations it remained closed, suggesting a stabilizing role for the Val side chain. The overlap of (m)GDV's Val with the carbon chain of pembrolizumab's Arg102 (*Figure 10—figure supplement 1*) and the similarity between the (m)GDV-induced PD-1 interface and the pembrolizumab–bound interface supports the 'capping' role of Arg102 in stabilizing the flat hydrophobic surface of PD-1. This mechanism is also consistent with models of macrocycle conformations generated by Balloon (*Vainio and Johnson, 2007*) docked to PD-1, which readily identify poses that align the mGDV

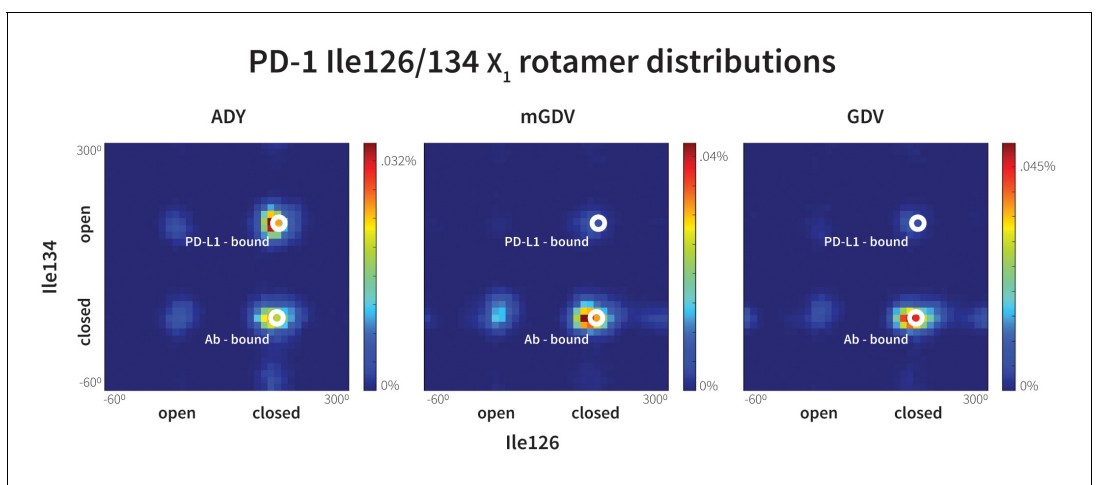

**Figure 11.** Macrocyclic mGDV motif induces structural changes in the PD-1 interface towards the pembrolizumab–bound state. Heat maps show the distributions of PD-1's Ile126 and Ile134 $X_1$ rotamer angles in MDs of the receptor interacting with the ADY PD-L1 trigger (left), the BMS macrocycle mGDV motif (center), and the GDV peptide. Data for each ligand were gathered from three distinct 200ns simulations. White dots on the plots indicate the rotamer angles of the same two residues in the pembrolizumab (Ab)–bound (*Horita et al., 2016*) and PD-L1–bound (*Zak et al., 2015*) cocrystal structures.

The following source data is available for figure 11:

**Source data 1.** Excel workbook with a single sheet containing the 2D histogram data for the heatmaps shown in *Figure 11*.

motif to corresponding PD-L1 and pembrolizumab interface residues (*Figure 10—figure supplements 1* and *2*), rationalizing the potency and specificity of the compound.

## Discussion

### Induced fit motif XDY shared by PD-1 ligands modulates the flexible PD-1 binding interface from hydrophilic to hydrophobic

Our studies show that apo PD-1 does not sample bound-like hydrophobic interface conformations, but instead presents a non-bound-like hydrophilic patch around Asn66 at the core of its binding interface (*Figure 2*). By mapping the effect of specific ligand contacts on the apo PD-1 interface, we identify a highly conserved subset of PD-L1/2 motifs responsible for coordinating Asn66 and triggering the transition from the hydrophilic to hydrophobic interface. Namely, Asp122/111 and the backbone O of PD-L1/2 Ala121/Trp110 form a robust, four-membered hydrogen bond network with Tyr68 and Asn66 that neutralizes the latter residue into a bound-like open state. Simultaneously, the conserved anchor Tyr123/112 stabilizes Ile134 into a bound-like state that, with Asn66 open, creates a hydrophobic surface that fluctuates between a patch and a cavity modulated by Ile126. These three linear ligand motifs (XDY), shared by both PD-L1/2, comprise the molecular key that unlocks the promiscuity of PD-1 by revealing a flexible hydrophobic binding surface (*Figure 4*).

### A single carbon atom difference can shift the hydrophobic PD-1 binding surface from a stable patch to a stable cavity

With XDY triggering the transition to the flexible hydrophobic surface, specificity toward the two PD-1 ligands is actualized by the formation of the hydrophobic patch when binding PD-L1 vs. the formation of hydrophobic cavity when binding PD-L2. These two states can be distinguished by the conformation of Ile126 (*Figure 2e*). For PD-L1, we show that the ADY motif is sufficient to stabilize the hydrophobic patch (*Figure 4c*). Specifically, the Ala121 C$\beta$ atom, which does not overlap with PD-1 apo NMR structures (*Figure 2d*), recruits Ile126 into the closed (patch) state. On the other hand, in the absence of C$\beta$, the GDY trigger stabilizes the open state of Ile126, producing a large hydrophobic interface cavity consistent with the pocket that buries PD-L2 Trp110. Note that the Trp in the WDY motif of PD-L2 readily fills the hydrophobic pocket as the XDY motif latches and opens Asn66 (*Figure 8*).

### Bound-like XDY residues and molecular recognition

The pre-arrangement of PD-L1/2 motifs XDY in bound-like conformations in the absence of the receptor is important for efficient ligand recognition and binding. Docking studies and peptide MDs highlight a critical role for the conserved Tyr123/112 anchor both in both molecular recognition and in modulating Ile134 during induced fit, both of which require the Tyr side chain to maintain a stable bound-like rotamer. Furthermore, simulations demonstrate that peptides such as GDG, mGDV, and GDV, which either lack or have a modified anchor analogue, cannot stabilize an open state of Ile126, highlighting an allosteric role for Tyr123/112 in splitting the PD-1 induced fit binding pathway.

Several anchors substitutes were tested in simulation starting in bound-like configurations similar to the cognate Tyr112/123. These MDs produced three broad types of PD-1 interface dynamics (*Figure 4—figure supplement 3*): (1) aromatic substitutions XDF and XDW stabilized either an open (X=G) or closed (X=A) pocket like the cognate XDY motif. (2) Polar substitutions XDH, XDR, and XDK were not accommodated in the hydrophobic anchor pocket and their side chains laid along the receptor surface, consistent with pembrolizumab's bound Arg102 (*Figure 10b*), producing a closed pocket like that of (m)GDV. (3) XDG or XDA resulted in open-closed fluctuations of both Ile134 and Ile126 (*Figure 4b,c*). These observations suggest that certain anchor mutations are tolerated by PD-1 and are consistent with mutagenesis studies showing that the Y112A PD-L2 point mutation slightly reduces, but does not abolish, binding to PD-1 (*Lázár-Molnár et al., 2008*). However, the observed conservation of Tyr123/112 in mammalian species (*Lázár-Molnár et al., 2008*) might suggest specific kinetic constraints on ligand recognition arising from hydrophobic contacts with Ile134 and the hydrogen bond with Glu136 (*Figure 9*), which are not shared by other sidechains.

In addition to the anchor residue, our peptide MDs also suggest an essential role for the conserved Asp122/111 in erecting a stable hydrogen bond network that opens PD-1 Asn66, which can

only be achieved by a bound-like Asp side chain. The primacy of these intermolecular interactions to PD-1 binding is reinforced by our MDs of apo PD-L1/2, which reveal that Tyr123/112 and Asp122/111 all remain in bound-like conformations in the absence of the receptor, primed to interact immediately upon interface association. Equally important is the fact that apo PD-1 structures all accommodate (i.e. do not block) any of contacts of the XDY scaffold, ensuring a rapid recognition process that facilitates subsequent induced fit transitions.

## Downhill binding pathways strongly suggest an induced fit binding mechanism

Our MDs demonstrate that the set of consecutive intermolecular interactions triggered by ADY and GDY peptides lead to energetically downhill binding pathways with no opposing energy barriers. These pathways strongly suggest that PD-1 occurs mostly by induced fit (*Figure 1*). Specifically, simulations and estimated $\Delta G_{open}$ values show that apo$_{BL}$ states of PD-1 are rare, which undermines a conformational selection mechanism. On the other hand, ligand-specific triggers are shown to efficiently shift the PD-1 interface conformational ensemble from a non-bound-like: bound-like ratio of roughly 44: 1 (in the apo ensemble) to roughly 1: 7 (in the encounter complex ensemble) (*Figure 7*). Unconstrained MDs of PD-L1/2 encounter complexes show that the geometry and chronology of triggering contacts is highly optimized, driving the transition from the non-bound-like to the bound-like states in less than 10 ns. This time scale promotes rapid recognition and ensures fast activation of this important T-cell checkpoint.

## Two-step binding pathway of PD-1 reveals a simple mechanism for selective promiscuity

Although regulatory proteins are promiscuous in that they bind multiple targets, they must also be specific so as to limit binding to just those targets. Our analysis of the binding mechanism to PD-1 reveals how these two seemingly contradictory requirements can be simultaneously achieved. Here, we show that apo PD-1 samples an ensemble of non-bound-like conformations that present an obstructive Asn66 on its interface, which likely prevents non-specific binding. The apo PDL1/2 interfaces feature a conserved, bound-like, XDY binding motif that holds the key to opening Asn66 and forming a flexible hydrophobic surface, which completes the first binding step. In the second step, the ligands then attune the flexible interface via specificity-determinant contacts (X=A for PD-L1, X=W for PD-L2) that modulate Ile126, splitting the binding pathway and stabilizing either a hydrophobic patch or a binding pocket (*Figures 2*, *4* and *7*). The key structural properties in this pathway are: (a) a flexible, non-bound-like apo receptor interface ensemble that presents an unfavorable binding surface, (b) a core subset of shared ligand binding motifs clustered about an anchor residue that latch the receptor interface but allow it to remain *flexible*, and (e) ligand-specific motifs that split the binding pathway by stabilizing different conformations of the flexible interface.

## Molecular triggers could be exploited to design small-molecule PD-1 antagonists

Bound cocrystal structures of the PD-1–targeting antibody pembrolizumab reveal that it exploits an evolutionarily designed induced fit trigger: the four-membered hydrogen bond network that opens Asn66 and makes the receptor interface hydrophobic. This same principle can be applied to design smaller molecular weight PD-1 inhibitors. We have shown that the mGDV motif of the Brystol-Myers-Squibb PD-1 inhibitor combines key pharmacophore features of both PD-L1 and pembrolizumab interfaces: the backbone O of the Gly resembles that of PD-L1's Ala121, the Asp side chain resembles PD-L1's Asp122, and the Val side chain resembles pembrolizumab's Arg102. Simulations suggest that this structural resemblance produces functionally similar dynamics by displacing receptor residue Asn66 (*Figure 10d*) and stabilizing a bound-like, flat hydrophobic surface formed by closed Ile126 and Ile134 (*Figures 7* and *11*). Docked conformations of the full inhibitor recapitulate most secondary native-like contacts in addition to the core triggering interactions (*Figure 10—figure supplements 1* and *2*). Taken together, these results support the idea that nature's mechanisms for modulating receptor surfaces might be exploited to design novel chemistries capable of transforming hard to drug targets into more druggable candidates.

## Selective promiscuity via induced fit offers potential advantages over conformational selection for multi-ligand regulatory proteins

Promiscuous regulatory proteins must optimize binding kinetics for multiple ligands by exploiting structural flexibility. Given nature's general mechanisms for flexibility-mediated binding (*Figure 1*), specificity toward multiple ligands could, in principle, be conferred either through conformational selection, by evolving the receptor to intrinsically sample different ligand-specific apo$_{BL}$ states, or by induced fit, by evolving interface interactions that efficiently drive transitions to the ligand-specific EC$_{BL}$ states. If conformational selection is used to achieve multi-ligand specificity, the binding pathway flux will de facto be limited by $\Delta G_{BL}^{apo}$, the free energy difference between each ligand-specific bound-like states and other states in the apo ensemble. In this scenario, a natural bottleneck would emerge as an increasing number of ligands would lead to lower association rates.

On the other hand, if selective promiscuity is conferred through induced fit, binding pathway flux will not depend on the fractional populations of apo ensemble microstates, but instead will be determined by the ligand-specific triggering mechanisms. We show here that induced fit can efficiently reshape the shallow polar interface of a flexible receptor into a hydrophobic interface amenable to binding multiple ligands by co-evolving a common set of intermolecular contacts. From an evolutionary perspective, this is an efficient approach to spawning novel protein interactions, since these core contacts can be designed just once. Selectivity to novel ligands can then be achieved by evolving relatively small sequence modifications around these core contacts. Perhaps more importantly, we note that contrary to conformational selection, the induced fit approach to selective promiscuity is in principle not limited by the total number of ligands.

It is interesting to note that many well-characterized eukaryotic regulatory domains (*Pawson and Scott, 1997*) bind to several linear binding sequences that share common motifs around an anchor residue and differ in other nearby regions. This trend suggests that the selective promiscuity via induced fit mechanism proposed here for PD-1 might apply elsewhere in nature. This possibility is currently being studied by analyzing the triggers of induced fit in other systems.

# Materials and methods

## Initial protein structures used in simulations

Molecular dynamics simulations (MDs) of the extracellular domain of PD-1 were run in triplicate using the first three solution NMR structures of apo human PD-1 (PDB ID: 2M2D [*Cheng et al., 2013*]). Before simulating specific receptor-ligand interactions, MDs of apo PD-1 were evaluated to ensure that the resultant dynamics are consistent with the experimentally derived apo NMR ensemble. As shown in *Figure 12*, apo MDs stabilize within about 2.0 Å backbone RMSD of their respective NMR starting points, suggesting that we can successfully sample native-like unbound states.

Available co-crystal structures of human PD-1/human PD-L1 (PDB ID: 4ZQK [*Zak et al., 2015*]), murine PD-1/human PD-L1 (PDB ID: 3BIK [*Lin et al., 2008*]) and murine PD-1/murine PD-L2 (PDB ID: 3BP5 [*Lázár-Molnár et al., 2008*]) complexes were used as templates for placement of peptides in bound-like loci at the receptor interface, and the dynamics of the PD-1 binding interface in response to interactions with different structural motifs on the ligands were analyzed. We focus on interactions relevant for the opening and closing of the pocket around Asn66. Based on co-crystals, we noticed that the core interacting residues of PD-L1 (Ala121, Asp122, Tyr123) and the homologous residues on PD-L2 (Trp110, Asp111, Tyr112) form critical hydrogen bonds (hydrogen bonds) shaping this pocket. Thus, to dissect the contribution of each contact, we simulate the effects of the receptor interacting with a diverse set of peptide derivatives of these specific ligand residues.

## Peptide ligand mimics used in simulations

Ten distinct PD-1 systems were simulated in order to dissect the ligand groups that trigger induced fit interface deformations on the receptor. These systems included the apo receptor in isolation and in complex with nine different peptides that mimic cognate ligand backbone and side chain interactions with the receptor (*Figure 3*): the anchor residue Tyr, the backbone peptide GGG, five peptides to probe role of ligand side chain contacts DY, GGY, GDG, ADG, GDY, the PD-L1 peptide ADY, and the mGDV peptide, which mimics a patented PD-1 inhibitor.

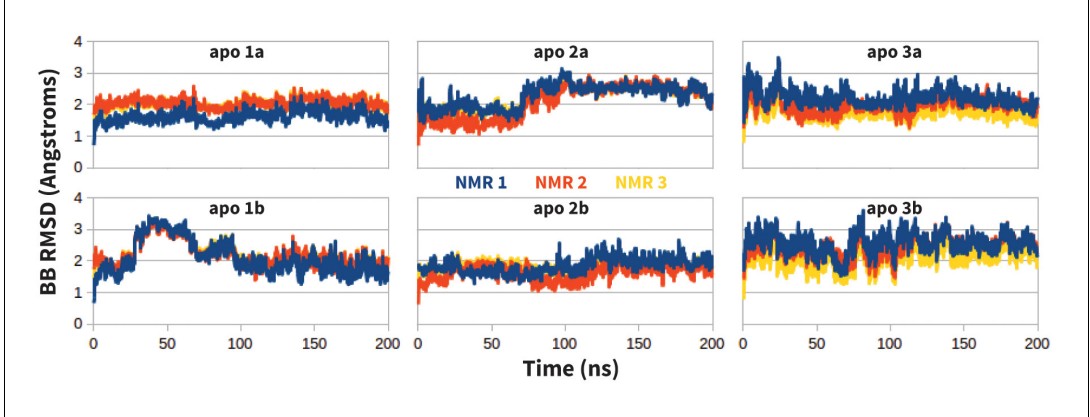

**Figure 12.** Stability of apo PD-1 simulations. Backbone RMSD of apo PD-1 to the first three NMR models (shown in blue, red, and yellow, respectively). Data are shown for six simulations: two replicates (**a,b**) starting from each of the first three NMR models (1,2,3).

The following source data is available for figure 12:

**Source data 1.** Excel workbook with a single sheet containing the time-series RMSD-to-unbound data from the apo PD-1 simulations plotted in *Figure 12*.

## Simulating PD-1–peptide interactions

To generate initial structures for our receptor-peptide MDs, NMR models 1–3 of the human PD-1 were backbone aligned to the murine receptor co-crystal (*Lin et al., 2008*) and peptides were modeled after the corresponding human PD-L1 interface residues Ala121–Tyr123, homologous to PD-L2 interface residues W110-Tyr112. Systems are simulated for 200 ns, resulting in three replicate MDs per system (including the apo PD-1 system, which does not include any peptide), and receptor interface dynamics are compared across systems to identify the ligand motifs and interactions responsible for structural transitions toward the bound-like receptor state. Harmonic restraints (100.0 kcal/mol) on all heavy atoms of ligand-mimicking peptides were used in simulation to prevent dissociation of the peptide from the receptor interface.

In peptide MDs, harmonic restraints (100.0 kcal/mol) were also placed on backbone atoms of non-interface PD-1 beta sheets residues 50–55, 80–81, 96–98, 106–109 and 120–122. These residues exhibit <0.35 Å backbone RMSD in the apo NMR ensemble, and previous studies have also shown that the conformational changes induced by ligand binding do not propagate through the major fold of PD-1 (*Cheng et al., 2013*). Hence, these restraints should not prevent our ability to sample the native-like binding dynamics of the receptor interface in biological conditions. The resolved portion of the N-terminal tail of PD-1 (residues 33–36), which in NMR models has <0.65 Å backbone RMSD, was also restrained so as to limit artificial mobility that might result from the fact that residues 1–32 were missing from simulation.

## Encounter complex modeling and simulation

Human PD-1–PD-L1 and PD-1–PD-L2 encounter complexes were modeled and then simulated in triplicate to probe induced fit trajectories and determine the chronology of inter-molecular interactions and specific interface deformations. We modeled encounter complexes by rigid body docking the extracellular domain of the apo receptor and the Ig-like V-type domains of the apo ligands, allowing no structural overlaps. Docked models of PD-L1 had an average backbone RMSD of 5.7 ± 1.2 to the human PD-1–PD-L1 cocrystal. Docked models of PD-L2 had an average backbone RMSD of 4.8 ± 1.8 to the murine PD-1–PD-L2 cocrystal (no human cocrystal is currently available for the PD-1–PD-L2 complex).

Structural models of apo human PD-L1 and PD-L2 that we used when building encounter complexes were generated by simulating the ligands in solution for 400 ns, using a VMD (*Humphrey et al., 1996*) clustering plugin (https://github.com/luisico/clustering) to cluster frames by backbone RMSD using a 3 Å cutoff, and taking the centroid frame of the largest cluster for each ligand. The initial structure for the PD-L1 clustering MDs was taken as the structure of the bound

human ligand from the co-crystal complex with murine PD-1 (PDB ID: 3BIK). As there are currently no available crystal structures of human PD-L2, a homology model was built as a starting point for the clustering simulation by manually mutating the bound structure of murine PD-L2 (PDB ID: 3BP5) and minimizing the resulting structure. We used the ClusPro protein-protein docking server (*Comeau et al., 2004*) to dock the top apo PD-L1 and PD-L2 centroid structures from their respective MDs to the first three NMR structures of apo human PD-1 (all three receptor structures are non-bound-like). Three bound-like candidate models for the PD-1–PD-L1/2 encounter complexes that correctly anchored Tyr123/112 were chosen from the ClusPro output. We then simulated these encounter complexes for 400 ns to probe the dynamics of the induced fit binding pathway.

## Simulation parameters

We ran MDs using AMBER14's (*Case et al., 2014*) pmemd.cuda module (*Götz et al., 2012*) and the AMBER ff12SB force field. The cutoff for non-bonded interactions was set at 10 Å. Systems were simulated in an octahedral TIP3P water box with periodic boundary conditions and a 12 Å buffer around the solute. Cl ions were added to the solvent to neutralize the charge of the systems. We minimized each system twice and then equilibrated them before beginning production runs. In the first minimization, solute atoms were held fixed through 500 steps of steepest descent and 500 steps of conjugate gradient minimization. In the second minimization, only the solute backbone atoms were restrained through 2000 steps of steepest descent and 3000 steps of conjugate gradient. After minimization, system temperatures were raised to 300 K over the course of a 200 ps constant volume simulation (with an integration step of 2 fs) during which the solute was fixed with weak (10.0 kcal/mol) restraints. Bonds involving hydrogens were held at constant length. For the production MDs, the 200–400 ns simulations were held at 300 K under constant pressure with the constraints as listed above for each system and an integration step size of 2 fs.

## Analysis tools

The PyMOL Molecular Graphics System v1.7.4.0 was used for structure preparation and analysis (*Schrödinger, 2010*). Trajectories were analyzed using VMDv1.9.2 (*Humphrey et al., 1996*) and the MDpocket software package v2.0 (*Le Guilloux et al., 2009*; *Schmidtke et al., 2010*) for cavity detection and volume/surface area measurement. Measurements of PD-1 binding pocket occlusion, shown in *Figures 2d* and *4d,e*, were calculated from molecular dynamics simulations of PD-1 using a Python script (available at https://github.com/npabon/md_pocket_occlusion; a copy is archived at https://github.com/elifesciences-publications/md_pocket_occlusion [*Pabon, 2016*]). Briefly, the script takes a molecular dynamics trajectory and a set of static reference atoms and identifies which reference atoms are overlapped in each frame of the simulation. Overlap occurs when any simulated atom crosses the 'clash radius' of a reference atom, the clash radius being equal to the sum of the van der Waals radii of the two atoms. The output of the script is the fractional occlusion of each reference atom position, equal to the percentage of simulation frames in which that reference atom is overlapped by simulated atoms. This script was used to evaluate the extent to which the Trp110 and Tyr112/123 binding cavities are open in simulations of PD-1 interaction with various peptides, simulations of apo PD-1, and the apo NMR ensemble of PD-1.

## Relative free energies of bound-like versus non-bound-like interfaces

We classified PD-1 interface conformations using two binary order parameters that define whether interface residues Asn66 and Ile126 are in their 'open' or 'closed' rotamer states. These parameters are used to distinguish the non-bound-like interface, where Asn66 is closed and Ile126 is open, from the PD-L1-specific bound-like state, where Asn66 is open and Ile126 is closed, and the PD-L2-specific bound-like state, where both Asn66 and Ile126 are open (*Figure 2e*). We estimated the energy differences $\Delta G_{BL}^{apo}$ and $\Delta G_{BL}^{EC}$ (*Figure 1*) using Maxwell-Boltzmann statistics by assessing the bound-like (BL) and non-bound-like (NBL) state population distributions in the apo and encounter complex (EC) receptor ensembles:

$$\frac{\left\langle n_{BL}^{apo/EC} \right\rangle}{\left\langle n_{NBL}^{apo/EC} \right\rangle} = e^{\frac{-\Delta G_{BL}^{apo/EC}}{k_B T}}$$

(1)

In the above equations, $\left\langle n_{BL}^{apo/EC} \right\rangle$ and $\left\langle n_{NBL}^{apo/EC} \right\rangle$ denote fractional equilibrium populations of the apo / encounter complex receptor ensembles in the bound-like and non-bound-like macrostates, and $k_BT$ is the product of the Boltzmann constant and temperature. We used MDs to generate the equilibrium ensembles of receptor conformations and analyzed the trajectories to calculate $\left\langle n_{BL/NBL}^{apo/EC} \right\rangle$ values.

MDs trajectories were analyzed as follows. Reference structures for the open and closed states of Asn66 were defined using its side chain configuration in the first apo NMR model and PD-L1-bound human cocrystal, respectively (Asn66 has <0.2 Å heavy atom RMSD between PD-L1 and PD-L2 cocrystals 4ZQK and 3BP5). Each frame of the MDs trajectory is labeled with the state to which the simulated Asn66 had the smaller side chain RMSD to the reference structure. Reference structures for the open and closed states of Ile126 were defined using its $X_1$ rotamer angle in the murine PD-L2 and human PD-L1 cocrystals, respectively, this angle being the main distinguishing feature between the two different ligand-bound interfaces (*Figure 2e*). Each frame of the MDs was labeled with the state to which the simulated Ile126 had the closest rotamer angle. The free energy changes of opening Asn66 and Ile126 are calculated using *Equation (1)* and then compared across different simulations in order to identify triggers of interface deformations.

## Acknowledgements

This work was funded by the National Institute of Health (grant GM097082 to CJC) and the National Science Foundation (grant 1247842 to NAP).

## Additional information

### Funding

| Funder | Grant reference number | Author |
|---|---|---|
| National Science Foundation | 1247842 | Nicolas A Pabon |
| National Institutes of Health | GM097082 | Carlos J Camacho |

The funders had no role in study design, data collection and interpretation, or the decision to submit the work for publication.

### Author contributions

NAP, Conceptualization, Data curation, Formal analysis, Funding acquisition, Investigation, Visualization, Methodology, Writing—original draft, Writing—review and editing; CJC, Conceptualization, Formal analysis, Supervision, Funding acquisition, Methodology, Writing—review and editing

### Author ORCIDs

Nicolas A Pabon, http://orcid.org/0000-0002-2591-4349
Carlos J Camacho, http://orcid.org/0000-0003-1741-8529

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
