## [Decision Letter]

Thank you for submitting your article "Probing protein flexibility reveals a mechanism for selective promiscuity" for consideration by *eLife*. Your article has been favorably evaluated by John Kuriyan (Senior Editor) and three reviewers, one of whom, Nir Ben-Tal (Reviewer #1), is a member of our Board of Reviewing Editors. The following individual involved in review of your submission has agreed to reveal their identity: Nikolay V Dokholyan (Reviewer #3).

The reviewers have discussed the reviews with one another and the Reviewing Editor has drafted the following remarks to elicit a response from you. Please consider the serious concerns in comments 1 and 2 below and respond with a plan of how you may be able to address these points by further theory or experiment, and indicate an approximate time from for the completion of such work. The editor and reviewers will evaluate your response and issue a binding recommendation.

Review:

The manuscript describes computational analyses of the ligand-responsiveness of the PD-1 immune checkpoint inhibitor. In spite of the availability of a crystal structure of bound PD-1, and NMR structures of the unbound (apo) protein, it is not understood how the receptor is able to bind more than one ligand with different binding modes but with specificity. The manuscript examines two possible pathways for ligand binding: conformational selection vs. induced fit. The binding dynamics is investigated using all-atom molecular dynamic simulations, where known NMR and co-crystal structures are used as the initial models for bound and unbound conformations. Various ligand-mimicking peptides are used to study the role of specific contacts between the protein and the ligand side chains. The conclusion is that binding follows the induced fit mechanism, which starts from the burying of the anchor motif of the ligand and follows by chronological sequence of pocket rearrangement in order to form and increase hydrophobic areas and form hydrogen bonds important for binding. The interesting aspect of this paper is the proposition that the initial docking of one or the other ligand triggers different conformational changes in the receptor that then ensures the appropriate specificity.

Given the enormous prominence of the checkpoint inhibitors as newly emerging treatments for a wide range of cancers, and in particular the efficacy of reagents capable of directing blockade of PD-1 associated processes, this report is potentially of wide spread interest and importance. Nevertheless, the reviewers and the editors have some serious reservations about the robustness of the results, and how general the principles are. We would like to give you the opportunity to revise the manuscript in order to address these reservations. You should note that these reservations are such that if they are not addressed adequately in the revision, then the paper may be deemed unsuitable for *eLife*. We note three essential points to address below. The first two refer to the most serious concerns, and will require additional computation, or perhaps even experiments. The third point is also essential, but we feel that you should be able to address it by revising the figures and accompanying text.

Essential issues to address:

1) While the computational results provide fertile ground for considering the detailed mechanisms underlying PD-L1 and PD-L2 recognition and potentially affinities, there are no prospective experimental studies to challenge and validate these interesting hypotheses. The lack of experimental support is further highlighted in the section describing the binding of a relatively high affinity (Kd ~1 nM) macrocycle developed by BMS. The PD-1 recognition surface assumed for this macrocycle is based on its sequence similarity with PD-L1, which is a reasonable hypothesis; however, there is no direct experimental support for this assumption in the current manuscript or anywhere in the literature. The reviewers are concerned that in its present form the paper does not present a reliable set of conclusions.

We recognize that your paper presents the results of a computational study, and that it may not be feasible for you to provide experimental data to support your concepts. In that case, please consider alternative ways to establish the robustness of your results.

For example, does PD-1 have more ligands beyond PD-L1 and PD-L2? "These three linear ligand motifs (XDY), shared by both PD-L1/2, comprise the molecular key[…]": Are there counter examples? That is, peptides that do not share the motif and were proved in experiments to not bind PD-1? If so, simulations with such negative controls should be added.

Other proteins in addition to PD-1 could be studied to examine how general the mechanism is.

2) Y123/112. Several tripeptides were used to examine the importance of this residue but these are just a small portion of the possible sample. How about other aromatic residues? Or HIS? Or other residues that could potentially support a hydrogen bond equivalent of OH_eta?

3) The illustrations in the paper need to be improved.

A diagram showing the intact receptor should be shown to provide context for the ligand interaction. In most figures the protein is shown using its molecular surface, without revealing the physicochemical nature and/or residues types underlying the surface. Thus, it is difficult to examine the fit of peptide binding. This is crucial since one of the main observations here is that the binding site shifts its polar nature upon binding. The current figures do not explain this clearly.

In Figure 2 it might be useful to zoom out a bit and show more of the binding pocket, so a reader can understand open/close residues orientation with respect to the pocket.

It might be useful to include the structures of the ligands themselves in Figure 3 for the comparison of mimicking motifs.

More information concerning the fractional overlap of atoms (Figure 4)) would be useful.

Other points to address:

1) "Interestingly, no small molecular weight inhibitor has been reported for this seemingly druggable interface cavity. This is likely due to our incomplete understanding of how PD-1's flexibility enables selectivity for two distinct ligand interfaces, only one of which stabilizes the hydrophobic pocket": Speculative. Perhaps nobody tried to target this interface? A reference should be added or the statement should be revised.

2) "PD-1 has proven to be a difficult target to disrupt using small molecules": A reference should be added or the statement should be deleted.

3) Figure 3. Should be GGY rather than GGG.

4) Overall, there are far too many abbreviations in this paper, and we ask that you use only the most essential abbreviations, and keep these to a minimum. In the current form, the paper is challenging to read. Why abbreviate molecular dynamics simulations as "MDS"? Why use "PPI" as an abbreviation? Please consider the ease with which a reader can follow what you are trying to say.

---

## [Author Response]

*Essential issues to address:*

*1) While the computational results provide fertile ground for considering the detailed mechanisms underlying PD-L1 and PD-L2 recognition and potentially affinities, there are no prospective experimental studies to challenge and validate these interesting hypotheses. The lack of experimental support is further highlighted in the section describing the binding of a relatively high affinity (Kd ~1 nM) macrocycle developed by BMS. The PD-1 recognition surface assumed for this macrocycle is based on its sequence similarity with PD-L1, which is a reasonable hypothesis; however, there is no direct experimental support for this assumption in the current manuscript or anywhere in the literature. The reviewers are concerned that in its present form the paper does not present a reliable set of conclusions.*

*We recognize that your paper presents the results of a computational study, and that it may not be feasible for you to provide experimental data to support your concepts. In that case, please consider alternative ways to establish the robustness of your results.*

*For example, does PD-1 have more ligands beyond PD-L1 and PD-L2? "These three linear ligand motifs (XDY), shared by both PD-L1/2, comprise the molecular key…": Are there counter examples? That is, peptides that do not share the motif and were proved in experiments to not bind PD-1? If so, simulations with such negative controls should be added.*

*Other proteins in addition to PD-1 could be studied to examine how general the mechanism is.*

The scarcity of experimental PD-1 binding data is unfortunate, especially given that in all likelihood this blockbuster therapeutic target has been subject to extensive biophysical analysis and drug discovery approaches. The incomplete picture of PD-1 binding may in part be due to the flexibility of the receptor, which we know creates essential challenges for structural analysis and small-molecule drug discovery [1]. The fact that so far only antibodies and large, complex macrocycles have been designed to bind PD-1 supports this claim. Furthermore, the fact that negative results from experiments probing PD-1 binding go unpublished prohibits us from using said data to constrain our models.

To our knowledge, the only currently available experimental data on PD-1 binding mechanics comes in the form of protein sequences, a handful of structures, and a small number of mutagenesis experiments that accompany them [2-8]. Our initial submission rationalized PD-L1/2 – bound structures of PD-1 as well as the sequence conservation of triggering residues and mutagenesis studies of PD-L1’s D111. In our revised submission, we rationalize the results of several additional mutagenesis experiments on murine PD-1 and PD-L2. For example, the fact that the K113A point mutation on PD-L2 abolishes binding to PD-1 can be explained by the fact that this residue is necessary to stabilize a bound-like configuration of PD-L2’s D111. We elaborate on this idea in the Results section titled “Conserved PD-L1/2 Asp122/111 form a specific intermolecular hydrogen bond network that opens PD-1 Asn66 and switches the receptor interface from hydrophilic to hydrophobic.”:

“The robust, four-membered hydrogen bond network between the Ala121/Trp110 backbone mimic, Asn66, Tyr68, and the Asp122/111 mimic that we observe in GDG MDs is fully consistent with all available structures and mutagenesis experiments. […] The importance of this stabilizing interaction is underscored by the fact that the K124S and K113A point mutations in PD-L1 and PD-L2, respectively, both abolish binding to PD-1 [4, 5].”

Additionally, a number PD-1 interface residues that we suggest are important for molecular recognition and/or induced fit have been shown to reduce or abolish binding when mutated to Ala [4]. This data is included in Table 4 and Table 5 in our revised submission.

Fortunately, since the time of our submission, several crystal structures have been added to the Protein Data Bank (5B8C, 5GGS, 5JXE) showing PD-1 bound to the antibody Pembrolizumab. These new experimental data have allowed us to, in our revised submission, prospectively validate the molecular triggers revealed by our molecular dynamics studies.

Pembrolizumab is one of the new blockbuster generation of emerging anti-cancer immunotherapies, and it binds to a PD-1 surface that shares the core of the PD-L1 and PD-L2 binding interfaces. Remarkably, even though the Ab shares no sequence similarity with PD-L1 and PD-L2, it participates in a hydrogen bond network with receptor residue Asn66 that is analogous to the network that we identified as a key modulator of PD-1’s selective promiscuity to its cognate ligands. This commonality between the two distinct binding mechanisms highlights the importance of this HB network as an evolutionary constraint on shifting the polar PD-1 binding interface into a mostly non-polar one. A new Results section titled has been added to our manuscript to provide a full structural analysis of the pembrolizumab – bound PD-1 interface in the context of our proposed PD-L1/2 binding mechanism. This section contains a new figure (Figure 10) that illustrates similarities between the pembrolizumab – bound PD-1 interface and the PD-L1 – bound PD-1 interface. The new section, titled “PD-1 – targeting antibody validates the critical role of Asn66 and suggests an anchor-independent binding mechanism with closed Ile126 and Ile134” is shown below:

“Recently, two FDA-approved PD-1 – targeting antibodies have emerged as part of a new generation of anti-cancer immune checkpoint inhibitors. […] Results of the GDG MDs thus rationalize the pembrolizumab binding mode and suggest an anchor-independent induced fit PD-1 binding pathway: one in which the antibody opens Asn66 using the canonical hydrogen bond network and stabilizes the resulting flat hydrophobic interface by ‘capping’ the closed states of Ile126/134 with the carbon chain of Arg102.”

With respect to the BMS macrocycle, we have de-emphasized our predicted binding models and removed them from the main figures, since to date we have no way of experimentally validating them. However, the PD-1 – bound cocrystals of pembrolizumab, which, similar to the BMS macrocycle does not have an anchor analogue, have allowed us to further explore possible macrocycle binding mechanisms. Specifically, we have added a section to the Results with a new figure showing that the macrocycle’s mGDV motif, aligned to pembrolizumab’s core interface residues, closes both Ile126 and Ile134 as in the pembrolizumab-bound cocrystals. The modified Results section, titled “Can molecular triggers be exploited to drug PD-1?”, is shown below:

“Although two PD-1 targeting antibodies already exist on the market, there are currently no FDA- approved small-molecule PD-1 inhibitors, despite enormous interest in this blockbuster immunotherapy target [23-25]. […] This mechanism is also consistent with models of macrocycle conformations generated by Balloon [38] docked to PD-1, which readily identify poses that align the mGDV motif to corresponding PD-L1 and pembrolizumab interface residues (Figure 10—figure supplement 1 and Figure 10—figure supplement 2), rationalizing the potency and specificity of the compound.”

To our knowledge, extracellular PD-1 has no additional cognate ligands beyond PD-L1 and PD-L2. Furthermore, no negative binding results have been reported for synthetic small molecules. The only available counter examples are the few experimentally tested PD-1 and PD-L1/2 point mutations (referenced above) that abolish or reduce ligand binding, all of which are rationalized by our results and are now included in the revised manuscript.

*2) Y123/112. Several tripeptides were used to examine the importance of this residue but these are just a small portion of the possible sample. How about other aromatic residues? Or HIS? Or other residues that could potentially support a hydrogen bond equivalent of OH_eta?*

Our primary aim of our manuscript is to identify the minimal set of native interactions responsible for PD- 1’s selective promiscuity via induced-fit, and the conservation of Tyr123/112 in both PD-L1 and PD-L2 across mammalian species [4] suggest a highly constrained role in this mechanism. As suggested by our reviewers, we investigated several additional peptides with different anchor substitutions and compared the resultant PD-1 interface deformations to those of the cognate peptides. These results are now included in Figure 4—figure supplement 3 and are discussed in the Discussion section titled “Bound-like XDY residues and molecular recognition” of our revised submission:

“The pre-arrangement of PD-L1/2 motifs XDY in bound-like conformations in the absence of the receptor is important for efficient ligand recognition and binding. Docking studies and peptide MDs highlight a critical role for the conserved Tyr123/112 anchor both in both molecular recognition and in modulating Ile134 during induced fit, both of which require the Tyr side chain to maintain a stable bound-like rotamer. […] However, the observed conservation of Tyr123/112 in mammalian species [4] might suggest specific kinetic constraints on ligand recognition arising from hydrophobic contacts with Ile134 and the hydrogen bond with Glu136 (Figure 9), which are not shared by other sidechains.”

3) The illustrations in the paper need to be improved.

*A diagram showing the intact receptor should be shown to provide context for the ligand interaction. In most figures the protein is shown using its molecular surface, without revealing the physicochemical nature and/or residues types underlying the surface. Thus, it is difficult to examine the fit of peptide binding. This is crucial since one of the main observations here is that the binding site shifts its polar nature upon binding. The current figures do not explain this clearly.*

Figure 2 has been modified and two figure supplements have been added to address this concern. Figure 2—figure supplement 1 shows a zoomed-out view of the intact cocrystal structures of the PD-1 – PD-L1 and PD-1 – PD-L2 interactions. The bound structures are shown side-by-side and identically oriented to facilitate comparison between the two ligands. Interface side chains on both the receptor and ligands are shown in order to provide a broad view of all intermolecular interactions, so as to provide context for the key triggering interactions described later in the paper. Other, related figure additions and modifications are discussed below.

The molecular surfaces in Figure 2 have been made transparent such that the underlying residues are visible, and the surfaces have been colored to represent their physiochemical properties. The subfigures have been zoomed-out to better depict intermolecular contacts and the overall fit of ligand binding. To further illustrate the induced-fit conformational changes of the PD-1 interface, the apo interface has been added as Figure 2, facilitating direct comparison between the unbound and ligand-bound interfaces in Figure 2. In Figure 2, the location of the core binding interface is indicated in order to clarify the polar-to-nonpolar shift. Similarly, the PD-1 molecular surface in Figure 10 has been made transparent and colored by physiochemical property in order to reveal the predicted inter-molecular interactions with the BMS macrocycle. In our revised submission, this figure was moved to Figure 10—figure supplement 1.

*In Figure 2 it might be useful to zoom out a bit and show more of the binding pocket, so a reader can understand open/close residues orientation with respect to the pocket.*

A new figure showing a zoomed-out view of the PD-1 binding pocket has been added as Figure 2—figure supplement 2. This supplement depicts the same structures as the initial Figure 2 but with additional detail, including several extra PD-1 interface side chains and the binding pocket volume of in each structure shown as transparent surfaces. This figure supplement better captures how the orientation PD-1 residues Asn66 and Ile126 regulate an open or closed binding pocket. Figure 2 of our original submission is left as is (though it moved to Figure 2 in our revised submission), since it is intended primarily to define the “open” vs. “closed” states of Asn66 and Ile126.

*It might be useful to include the structures of the ligands themselves in Figure 3 for the comparison of mimicking motifs.*

As suggested, we have added the core PD-L1 and PD-L2 interface residues to Figure 3 to provide additional context for the mimicking motifs.

*More information concerning the fractional overlap of atoms (Figure 4)) would be useful.*

We have switched our terminology from ‘fractional overlap’ to ‘fractional occlusion’ to clarify that we are measuring the extent to which the PD-1 interface cavities are obstructed by the closed-pocket conformations of the receptor interface residues. The Analysis Tools subsection of our Materials and methods has been expanded in order to better explain the procedure by which we calculate fractional occlusion:

“Measurements of PD-1 binding pocket occlusion, shown in Figure 2 and Figure 4, were calculated from molecular dynamics simulations of PD-1 using a Python script (https://github.com/npabon/md_pocket_occlusion). […] This script was used to evaluate the extent to which the Trp110 and Tyr112/123 binding cavities are open in simulations of PD-1 interaction with various peptides, simulations of apo PD-1, and the apo NMR ensemble of PD-1.”

Additionally, brief one-sentence summaries of this procedure have been added to the captions for Figure 2 and Figure 4, along with a reference to Materials and methods for complete details:

“Figure 2. Flexibility of the PD-1 binding interface[…] (d) Fractional occlusion of each bound-like Trp110 and Tyr123/112 atom position in the NMR ensemble of apo PD-1. Numerical values at each atom position denote the fraction of NMR frames that overlap, or “occlude”, that position (see Materials and methods for full details of how fractional occlusion is calculated). Aside from the Cβ, the Trp110 pocket is mostly occluded in the apo PD-1 ensemble, whereas the Tyr123/112 anchor pocket is largely open[…]”

“Figure 4. Dynamics of PD-1 binding interface in the presence of different ligands[…] (d) Fractional occlusion of each bound-like Trp110 atom position in simulations of PD-1 interacting with the GDY peptide show an open Trp110 binding pocket. The fractional occlusion of a Trp110 atom position is defined as the percentage of simulation frames in which a PD-1 atom overlaps, or “occludes”, that position (see Materials and methods for full details of how fractional occlusion is calculated). (e) Fractional occlusion of each bound-like Trp110 atom position in simulations of PD-1 interacting with the ADY peptide show a closed Trp110 binding pocket.”

*Other points to address:*

*1) "Interestingly, no small molecular weight inhibitor has been reported for this seemingly druggable interface cavity. This is likely due to our incomplete understanding of how PD-1's flexibility enables selectivity for two distinct ligand interfaces, only one of which stabilizes the hydrophobic pocket": Speculative. Perhaps nobody tried to target this interface? A reference should be added or the statement should be revised.*

Target flexibility remains an essential challenge for structure-based drug discovery [1]. While we cannot be sure that PD-1’s flexibility is the primary reason for the absence of small-molecule PD-1 inhibitors, we can be confident that this absence is not for lack of trying. It is clear that a blockbuster target like PD-1 would be on the radar of pharma, biotech, and academic labs [10-12]. Indeed, my lab, and that of my collaborator Dr. Alexander Dömling, Chair of Drug Design at the University of Groningen, made several attempts to design PD-1 inhibitors that bind to the hydrophobic cavity observed in the PD-1 – PD-L2 cocrystal. Dr. Dömling’s lab produced the first co-crystal of human PD-1 bound to human PD-L1 [3], and also designed small molecules that bind PD-L1 [13], but they so far have no validated hits against PD-1. On this point, we note that the complexity of the patented BMS PD-1 macrocyclic inhibitors is clear evidence that significant efforts has been devoted in pharma towards this important target. For clarification, we have revised the statement in question to read as follows:

“To date, no small molecular weight PD-1 inhibitors have been reported in the literature despite the importance of this blockbuster target [10-12]. […] It is reasonable to assume that the flexibility of the Trp110 pocket, and the fact that in the apo state it is largely occluded by the unmatched, polar NH2 group of Asn66 (Figure 2), would present significant obstacles to traditional structure-based drug-design methods attempting to target this cavity [1].”

*2) "PD-1 has proven to be a difficult target to disrupt using small molecules": A reference should be added or the statement should be deleted.*

As above, references have been added and the statement has been revised as follows:

“Although two PD-1 targeting antibodies already exist on the market, there are no small-molecule PD-1 inhibitors in clinical trial, despite the enormous interest in this blockbuster immunotherapy target [10-12].”

*3) Figure 3. Should be GGY rather than GGG.*

The label for the GGY peptide has been corrected.

*4) Overall, there are far too many abbreviations in this paper, and we ask that you use only the most essential abbreviations, and keep these to a minimum. In the current form, the paper is challenging to read. Why abbreviate molecular dynamics simulations as "MDS"? Why use "PPI" as an abbreviation? Please consider the ease with which a reader can follow what you are trying to say.*

The abbreviations for hydrogen bond (HB) and protein-protein interaction (PPI) have been removed from the paper. Abbreviations for bound-like (BL), non-bound-like (NBL), and encounter complex (EC) have been removed everywhere except where they refer to the specific characteristic states shown in Figure 1, or the free energy differences between these states, such as in Equation (43). Our original abbreviation for molecular dynamics simulation (MDS) has been changed to MDs following the convention often found in the literature.

References

1) Cozzini P, Kellogg GE, Spyrakis F, Abraham DJ, Costantino G, Emerson A, Fanelli F, Gohlke H, Kuhn LA, Morris GM, Orozco M, Pertinhez TA, Rizzi M, Sotriffer CA. Target flexibility: an emerging consideration in drug discovery and design. J Med Chem. 2008;51(16):6237-55. Epub 2008/09/13. doi: 10.1021/jm800562d. PubMed PMID: 18785728; PubMed Central PMCID: PMC2701403.

2) Cheng X, Veverka V, Radhakrishnan A, Waters LC, Muskett FW, Morgan SH, Huo J, Yu C, Evans EJ, Leslie AJ, Griffiths M, Stubberfield C, Griffin R, Henry AJ, Jansson A, Ladbury JE, Ikemizu S, Carr MD, Davis SJ. Structure and interactions of the human programmed cell death 1 receptor. J Biol Chem. 2013;288(40):11771-85. Epub 2013/02/19. doi: 10.1074/jbc.M112.448126. PubMed PMID: 23417675; PubMed Central PMCID: PMC3636866.

3) Zak KM, Kitel R, Przetocka S, Golik P, Guzik K, Musielak B, Domling A, Dubin G, Holak TA. Structure of the Complex of Human Programmed Death 1, PD-1, and Its Ligand PD-L1. Structure. 2015;23(6):2341-8. doi: 10.1016/j.str.2015.09.010. PubMed PMID: 26602187.

4) Lazar-Molnar E, Yan Q, Cao E, Ramagopal U, Nathenson SG, Almo SC. Crystal structure of the complex between programmed death-1 (PD-1) and its ligand PD-L2. Proc Natl Acad Sci U S A. 2008;105(10):10483-8. Epub 2008/07/22. doi: 10.1073/pnas.0804453105. PubMed PMID: 18641123; PubMed Central PMCID: PMC2492495.

5) Lin DY, Tanaka Y, Iwasaki M, Gittis AG, Su HP, Mikami B, Okazaki T, Honjo T, Minato N, Garboczi DN. The PD-1/PD-L1 complex resembles the antigen-binding Fv domains of antibodies and T cell receptors. Proc Natl Acad Sci U S A. 2008;105(19):3011-6. Epub 2008/02/22. doi: 10.1073/pnas.0712278105. PubMed PMID: 18287011; PubMed Central PMCID: PMC2268576.

6) Horita S, Nomura Y, Sato Y, Shimamura T, Iwata S, Nomura N. High-resolution crystal structure of the therapeutic antibody pembrolizumab bound to the human PD-1. Sci Rep. 2016;6:35297. doi: 10.1038/srep35297. PubMed PMID: 27734966; PubMed Central PMCID: PMCPMC5062252.

7) Lee JY, Lee HT, Shin W, Chae J, Choi J, Kim SH, Lim H, Won Heo T, Park KY, Lee YJ, Ryu SE, Son JY, Lee JU, Heo YS. Structural basis of checkpoint blockade by monoclonal antibodies in cancer immunotherapy. Nat Commun. 2016;7:13354. doi: 10.1038/ncomms13354. PubMed PMID: 27796306; PubMed Central PMCID: PMCPMC5095608.

8) Na Z, Yeo SP, Bharath SR, Bowler MW, Balikci E, Wang CI, Song H. Structural basis for blocking PD-1-mediated immune suppression by therapeutic antibody pembrolizumab. Cell Res. 2017;27(43):147-50. doi: 10.1038/cr.2016.77. PubMed PMID: 27325296; PubMed Central PMCID: PMCPMC5223238.

9) Champ PC, Camacho CJ. FastContact: a free energy scoring tool for protein-protein complex structures. Nucleic Acids Res. 2007;35(Web Server issue):W556-60. Epub 2007/06/01. doi: 10.1093/nar/gkm326. PubMed PMID: 17537824; PubMed Central PMCID: PMC1933237.

10) Zarganes-Tzitzikas T, Konstantinidou M, Gao Y, Krzemien D, Zak K, Dubin G, Holak TA, Domling A. Inhibitors of programmed cell death 1 (PD-1): a patent review (2010-2015). Expert Opin Ther Pat. 2016;26(38):973-7. doi: 10.1080/13543776.2016.1206527. PubMed PMID: 27367741.

11) Domling A, Holak TA. Programmed death-1: therapeutic success after more than 100 years of cancer immunotherapy. Angew Chem Int Ed Engl. 2014;53(38):2286-8. Epub 2014/01/30. doi: 10.1002/anie.201307906. PubMed PMID: 24474145.

12) Couzin-Frankel J. Breakthrough of the year 2013. Cancer immunotherapy. Science. 2013;342(6165):1432-3. doi: 10.1126/science.342.6165.1432. PubMed PMID: 24357284.

13) Zak KM, Grudnik P, Guzik K, Zieba BJ, Musielak B, Domling A, Dubin G, Holak TA. Structural basis for small molecule targeting of the programmed death ligand 1 (PD-L1). Oncotarget. 2016;7(14):30323-35. doi: 10.18632/oncotarget.8730. PubMed PMID: 27083005; PubMed Central PMCID: PMCPMC5058683.

14) Laskowski RA, Luscombe NM, Swindells MB, Thornton JM. Protein clefts in molecular recognition and function. Protein Sci. 1996;5(6):2438-52. doi: 10.1002/pro.5560051206. PubMed PMID: 8976552; PubMed Central PMCID: PMCPMC2143314.

15) Liang J, Edelsbrunner H, Woodward C. Anatomy of protein pockets and cavities: measurement of binding site geometry and implications for ligand design. Protein Sci. 1998;7(38):1884-97. doi: 10.1002/pro.5560070905. PubMed PMID: 9761470; PubMed Central PMCID: PMCPMC2144175.

16) Cheng AC, Coleman RG, Smyth KT, Cao Q, Soulard P, Caffrey DR, Salzberg AC, Huang ES. Structure-based maximal affinity model predicts small-molecule druggability. Nat Biotechnol. 2007;25(43):71-5. doi: 10.1038/nbt1273. PubMed PMID: 17211405.